# Test-Time Adaptation to Distribution Shift by Confidence Maximization and Input Transformation

## Abstract

Deep neural networks often exhibit poor performance on data that is unlikely under the train-time data distribution, for instance data affected by corruptions. Previous works demonstrate that test-time adaptation to data shift, for instance using entropy minimization, effectively improves performance on such shifted distributions. This paper focuses on the fully test-time adaptation setting, where only unlabeled data from the target distribution is required. This allows adapting arbitrary pretrained networks. Specifically, we propose a novel loss that improves test-time adaptation by addressing both premature convergence and instability of entropy minimization. This is achieved by replacing the entropy by a non-saturating surrogate and adding a diversity regularizer based on batch-wise entropy maximization that prevents convergence to trivial collapsed solutions. Moreover, we propose to prepend an input transformation module to the network that can partially undo test-time distribution shifts. Surprisingly, this preprocessing can be learned solely using the fully test-time adaptation loss in an end-to-end fashion without any target domain labels or source domain data. We show that our approach outperforms previous work in improving the robustness of publicly available pretrained image classifiers to common corruptions on such challenging benchmarks as ImageNet-C.

## 1 Introduction

Deep neural networks achieve impressive performance on test data, which has the same distribution as the training data. Nevertheless, they often exhibit a large performance drop on test (target) data which differs from training (source) data; this effect is known as data shift (Quionero-Candela et al., 2009) and can be caused for instance by image corruptions. There exist different methods to improve the robustness of the model during training (Geirhos et al., 2019; Hendrycks et al., 2019; Tzeng et al., 2017). However, generalization to different data shifts is limited since it is infeasible to include sufficiently many augmentations during training to cover the excessively wide range of potential data shifts (Mintun et al., 2021a). Alternatively, in order to generalize to the data shift at hand, the model can be adapted during test-time. Unsupervised domain adaptation methods such as Vu et al. (2019) use both source and target data to improve the model performance during test-time. In general source data might not be available during inference time, e.g., due to legal constraints (privacy or profit). Therefore we focus on the *fully test-time adaptation* setting (Wang et al., 2020): model is adapted to the target data during test time given only the arbitrarily pretrained model parameters and unlabeled target data that share the same label space as source data. We extend the work of Wang et al. (2020) by introducing a novel loss function, using a diversity regularizer, and prepending a parametrized input transformation module to the network. We show that our approach outperform previous works and make pretrained models robust against common corruptions on image classification benchmarks as ImageNet-C (Hendrycks & Dietterich, 2019) and ImageNet-R (Hendrycks et al., 2020).

Sun et al. (2020) investigate test-time adaptation using a self-supervision task. Wang et al. (2020) and Liang et al. (2020) use the entropy minimization loss that uses maximization of prediction confidence as self-supervision signal during test-time adaptation. Wang et al. (2020) has shown that such loss performs better adaptation than a proxy task (Sun et al., 2020). When using entropy minimization, however, high confidence predictions do not contribute to the loss significantly anymore and thus provide little self-supervision. This is a drawback since high-confidence samples provide the most

trustworthy self-supervision. We mitigate this by introducing two novel loss functions that ensure that gradients of samples with high confidence predictions do not vanish and learning based on self-supervision from these samples continues. Our losses do not focus on minimizing entropy but on minimizing the *negative log likelihood ratio* between classes; the two variants differ in using either soft or hard pseudo-labels. In contrast to entropy minimization, the proposed loss functions provide non-saturating gradients, even when there are high confident predictions. Figure 1 provides illustration of the losses and the resulting gradients. Using these new loss functions, we are able to improve the network performance under data shifts in both online and offline adaptation settings.

In general, self-supervision by confidence maximization can lead to collapsed trivial solutions, which make the network to predict only a single or a set of classes independent of the input. To overcome this issue a *diversity regularizer* (Liang et al., 2020; Wu et al., 2020) can be used, that acts on a batch of samples. It encourages the network to make diverse class predictions on different samples. We extend the regularizer by including a moving average, in order to include the history of the previous batches and show that this stabilizes the adaptation of the network to unlabeled test samples. Furthermore we also introduce a parametrized *input transformation module*, which we prepend to the network. The module is trained in a fully test-time adaptation manner using the proposed loss function, and without using source data or target labels. It aims to partially undo the data shift at hand and helps to further improve the performance on image classification benchmark with corruptions.

Since our method does not change the training process, it allows to use any pretrained models. This is beneficial because any good performing pretrained network can be readily reused, e.g., a network trained on some proprietary data not available to the public. We show, that our method significantly improves performance of different pretrained models that are trained on clean ImageNet data.

In summary our main contributions are as follows: we propose non-saturating losses based on the negative log likelihood ratio, such that gradients from high confidence predictions still contribute to test-time adaptation. We extend diversity regularizer to its moving average to include the history of previous batch samples to prevent the model collapsing to trivial solutions. We also introduce an input transformation module, which partially undoes the data shift at hand. We show that the performance of different pretrained models can be significantly improved on ImageNet-C and ImageNet-R.

## 2 RELATED WORK

**Common image corruptions** are potentially stochastic image transformations motivated by real-world effects that can be used for evaluating a model's robustness. One such benchmark, ImageNet-C (Hendrycks & Dietterich, 2019), contains simulated corruptions such as noise, blur, weather effects, and digital image transformations. Additionally, Hendrycks et al. (2020) proposed three data sets containing real-world distribution shifts, including Imagenet-R. Most proposals for improving robustness involve special training protocols, requiring time and additional resources. This includes data augmentation like Gaussian noise (Ford et al., 2019; Lopes et al., 2019; Hendrycks et al., 2020), CutMix (Yun et al., 2019), AugMix (Hendrycks et al., 2019), training on stylized images (Geirhos et al., 2019; Kamann et al., 2020) or against adversarial noise distributions (Rusak et al., 2020a). Mintun et al. (2021b) pointed out that many improvements on ImageNet-C are due to data augmentations which are too similar to the test corruptions, that is: overfitting to ImageNet-C occurs. Thus, the model might be less robust to corruptions not included in the test set of ImageNet-C.

**Unsupervised domain adaptation** methods train a joint model of source and target domain by cross-domain losses to find more general and robust features, e. g. optimize feature alignment (Quiñonero-Candela et al., 2008; Sun et al., 2017) between domains, adversarial invariance (Ganin & Lempitsky, 2015; Tzeng et al., 2017; Ganin et al., 2016; Hoffman et al., 2018), shared proxy tasks (Sun et al., 2019) or adapt entropy minimization via an adversarial loss (Vu et al., 2019). While these approaches are effective, they require explicit access to source and target data at the same time, which may not always be feasible. Our approach works with any pretrained model and only needs target data.

**Test-time adaptation** is a setting, when training (source) data is unavailable at test-time. It is related to *source free adaptation*, where several works use generative models, alter training (Kundu et al., 2020; Li et al., 2020b; Kurmi et al., 2021; Yeh et al., 2021) and require several thousand epochs to adapt to the target data (Li et al., 2020b; Yeh et al., 2021). Besides, there is another line of work (Sun et al., 2020; Schneider et al., 2020; Nado et al., 2021; Benz et al., 2021; Wang et al., 2020) that

interpret the common corruptions as data shift and aim to improve the model robustness against these corruptions with efficient test-time adaptation strategy to facilitate online adaptation. such settings spare the cost of additional computational overhead. Our work also falls in this line of research and aims to adapt the model to common corruptions efficiently with both online and offline adaptation.

Sun et al. (2020) update feature extractor parameters at test-time via a self-supervised proxy task (predicting image rotations). However, Sun et al. (2020) alter the training procedure by including the proxy loss into the optimization objective as well, hence arbitrary pretrained models cannot be used directly for test-time adaptation. Inspired by the domain adaptation strategies (Maria Carlucci et al., 2017; Li et al., 2016), several works (Schneider et al., 2020; Nado et al., 2021; Benz et al., 2021) replace the estimates of Batch Normalization (BN) activation statistics with the statistics of the corrupted test images. Fully test time adaptation, studied by Wang et al. (2020) (TENT) uses entropy minimization to update the channel-wise affine parameters of BN layers on corrupted data along with the batch statistics estimates. SHOT (Liang et al., 2020) also uses entropy minimization and a diversity regularizer to avoid collapsed solutions. SHOT modifies the model from the standard setting by adopting weight normalization at the fully connected classifier layer during training to facilitate their pseudo labeling technique. Hence, SHOT is not readily applicable to arbitrary pretrained models.

We show that pure entropy minimization (Wang et al., 2020; Liang et al., 2020) as well as alternatives such as max square loss (Chen et al., 2019) and Charbonnier penalty (Yang & Soatto, 2020) results in vanishing gradients for high confidence predictions, thus inhibiting learning. Our work addresses this issue by proposing a novel non-saturating loss, that provides non-vanishing gradients for high confidence predictions. We show that our proposed loss function improves the network performance through test-time adaptation. In particular, performance on corruptions of higher severity improves significantly. Furthermore, we add and extend the diversity regularizer (Liang et al., 2020; Wu et al., 2020) to avoid collapse to trivial, high confidence solutions. Existing diversity regularizers (Liang et al., 2020; Wu et al., 2020) act on a batch of samples, hence the number of classes has to be smaller than the batch size. We mitigate this problem by extending the regularizer to a moving average version. Li et al. (2020a) also use a moving average to estimate the entropy of the unconditional class distribution but source data is used to estimate the gradient of the entropy. In contrast, our work does not need access to the source data since the gradient is estimated using only target data. Prior work Tzeng et al. (2017); Rusak et al. (2020b); Talebi & Milanfar (2021) transformed inputs by an additional module to overcome domain shift, obtain robust models, and also to learn to resize. In our work, we prepend an input transformation module to the model, but in contrast to former works, this module is trained purely at test-time to partially undo the data shift at hand to aid the adaptation.

## 3 METHOD

We propose a novel method for fully test-time adaption. We assume that a neural network $f_\theta$ with parameters $\theta$ is available that was trained on data from some distribution $\mathcal{D}$, as well a set of (unlabeled) samples $X \sim \mathcal{D}'$ from a target distribution $\mathcal{D}' \neq \mathcal{D}$ (importantly, no samples from $\mathcal{D}$ are required). We frame fully test-time adaption as a two-step process: (i) Generate a novel network $g_\phi$ based on $f_\theta$, where $\phi$ denotes the parameters that are adapted. A simple variant for this is $g = f$ and $\phi \subseteq \theta$ Wang et al. (2020). However, we propose a more expressive and flexible variant in Section 3.1. (ii) Adapt the parameters $\phi$ of $g$ on $X$ using an unsupervised loss function $L$. We propose two novel losses $L_{slr}$ and $L_{hlr}$ in Section 3.2 that have non-vanishing gradients for high-confidence self-supervision.

### 3.1 INPUT TRANSFORMATION

We propose to define the adaptable model as $g = f \circ d$. That is: we preprend a trainable network $d$ to $f$. The motivation for the additional component $d$ is to increase expressivity of $g$ such that it can learn to (partially) undo the domain shift $\mathcal{D} \to \mathcal{D}'$. Specifically, we choose $d(x) = \gamma \cdot [\tau x + (1-\tau)r_\psi(x)] + \beta$, where $\tau \in \mathbb{R}$, $(\beta, \gamma) \in \mathbb{R}^{n_{in}}$ with $n_{in}$ being the number of input channels, $r_\psi$ being a network with identical input and output shape, and $\cdot$ denoting elementwise multiplication. Specifically, $\beta$ and $\gamma$ implement a channel-wise affine transformation and $\tau$ implements a convex combination of unchanged input and the transformed input $r_\psi(x)$. By choosing $\tau = 1$, $\gamma = \mathbf{1}$, $\beta = \mathbf{0}$, we ensure $d(x) = x$ and thus $g = f$ at initialization. In principle, $r_\psi$ can be chosen arbitrarily. Here, we choose $r_\psi$ as a simple stack of $3 \times 3$ convolutions, group normalization, and ReLUs (refer Sec. A.2 for details). However, exploring other choices would be an interesting avenue for future work.

Importantly, while the motivation for $d$ is to learn to partially undo a domain shift $\mathcal{D} \to \mathcal{D}'$, we train $d$ end-to-end in the fully test-time adaptation setting on data $X \sim \mathcal{D}'$, without any access to samples from the source domain $\mathcal{D}$, based on the losses proposed in Section 3.2. The modulation parameters of $g_\phi$ are $\phi = (\beta, \gamma, \tau, \psi, \theta')$, where $\theta' \subseteq \theta$. That is, we adapt only a subset of the parameters $\theta$ of the pretrained network $f$. We largely follow Wang et al. (2020) in adapting only the affine parameters of normalization layers in $f$ while keeping parameters of convolutional kernels unchanged. Additionally, batch normalization statistics (if any) are adapted to the target distribution.

Note that the proposed method is applicable to any pretrained network that contains normalization layers with a channel-wise affine transformation. For networks with no affine transformation layers, one can add such layers into $f$ that are initialized to identity as part of model augmentation.

## 3.2 ADAPTATION OBJECTIVE

We propose a loss function $L = L_{\text{div}} + \delta L_{\text{conf}}$ for fully test-time network adaptation that consists of two components: (i) a term $L_{\text{div}}$ that encourages predictions of the network over the adaptation dataset $X$ that match a target distribution $p_{\mathcal{D}'}(y)$. This can help avoiding test-time adaptation collapsing to too narrow distributions such as always predicting the same or very few classes. If $p_{\mathcal{D}'}(y)$ is (close to) uniform, it acts as a diversity regularizer. (ii) A term $L_{\text{conf}}$ that encourages high confidence prediction on individual datapoints. We note that test-time entropy minimization (TENT) (Wang et al., 2020) fits into this framework by choosing $L_{\text{div}} = 0$ and $L_{\text{conf}}$ as the entropy.

### 3.2.1 CLASS DISTRIBUTION MATCHING $L_{div}$

Assuming knowledge of the class distribution $p_{\mathcal{D}'}(y)$ on the target domain $\mathcal{D}'$, we propose to add a term to the loss that encourages the empirical distribution of (soft) predictions of $g_\phi$ on $X$ to match this distribution. Specifically, let $\hat{p}_{g_\phi}(y)$ be an estimate of the distribution of (soft) predictions of $g_\phi$. We use the Kullback-Leibler divergence $L_{\text{div}} = D_{KL}(\hat{p}_{g_\phi}(y) \,||\, p_{\mathcal{D}'}(y))$ as loss term. In some applications information about the target class distribution is available, e.g. in medical data it might be known that there is a large class imbalance. In general this information is not available, and here we assume a uniform distribution of $p_{\mathcal{D}'}(y)$, which corresponds to maximizing the entropy $H(\hat{p}_{g_\phi}(y))$. Similar assumption has been made in SHOT to circumvent the collapsed solutions.

Since the estimate $\hat{p}_{g_\phi}(y)$ depends on $\phi$, which is continuously adapted, it needs to be re-estimated on a per-batch level. Since re-estimating $\hat{p}_{g_\phi}(y)$ from scratch would be computational expensive, we propose to use a running estimate that tracks the changes of $\phi$ as follows: let $p_{t-1}(y)$ be the estimate at iteration $t-1$ and $p_t^{emp} = \frac{1}{n} \sum_{k=1}^{n} \hat{y}^{(k)}$, where $\hat{y}^{(k)}$ are the predictions (confidences) of $g_\phi$ on a mini-batch of $n$ inputs $x^{(k)} \sim X$. We update the running estimate via $p_t(y) = \kappa \cdot \text{sg}(p_{t-1}(y)) + (1 - \kappa) \cdot p_t^{emp}$, where sg refers stop-gradient. The loss becomes $L_{\text{div}} = D_{KL}(p_t(y) \,||\, p_{\mathcal{D}'}(y))$ accordingly. Unlike Li et al. (2020a), our approach only requires target but no source data to estimate the gradient.

### 3.2.2 CONFIDENCE MAXIMIZATION $L_{conf}$

We motivate our choice of $L_{\text{conf}}$ step-by-step from the (unavailable) supervised cross-entropy loss: for this, let $\hat{y} = g_\phi(x)$ be the predictions (confidences) of model $g_\phi$ and $H(\hat{y}, y^r) = -\sum_c y_c^r \log \hat{y}_c$ be the cross-entropy between prediction $\hat{y}$ and some reference $y^r$. Let the last layer of $g$ be a softmax activation layer softmax. That is $\hat{y} = \text{softmax}(o)$, where $o$ are the network's logits. We can rewrite the cross-entropy in terms of the logits $o$ and a one-hot reference $y^r$ as follows: $H(\text{softmax}(o), y^r) = -o_{c^r} + \log \sum_{i=1}^{n_{cl}} e^{o_i}$ where $c^r$ is the index of the 1 in $y^r$ and $n_{cl}$ is the number of classes.

When labels being available for the target domain (which we do not assume) in the form of a one-hot encoded reference $y_t$ for data $x_t$, one could use the *supervised cross-entropy loss* by setting $y^r = y_t$ and using $L_{sup}(\hat{y}, y^r) = H(\hat{y}, y^r) = H(\hat{y}, y_t)$. Since fully test-time adaptation assumes no label information, supervised cross-entropy loss is not applicable and other options for $y^r$ need to be used.

One option is (hard) *pseudo-labels*. That is, one defines the reference $y^r$ based on the network predictions $\hat{y}$ via $y^r = \text{onehot}(\hat{y})$, where onehot creates a one-hot reference with the 1 corresponding to the class with maximal confidence in $\hat{y}$. This results in $L_{pl}(\hat{y}) = H(\hat{y}, \text{onehot}(\hat{y})) = -\log \hat{y}_{c^*}$, with $c^* = \arg \max \hat{y}$. One disadvantage with this loss is that the (hard) pseudo-labels ignore uncertainty in the network predictions during self-supervision. This results in large gradient magnitudes with

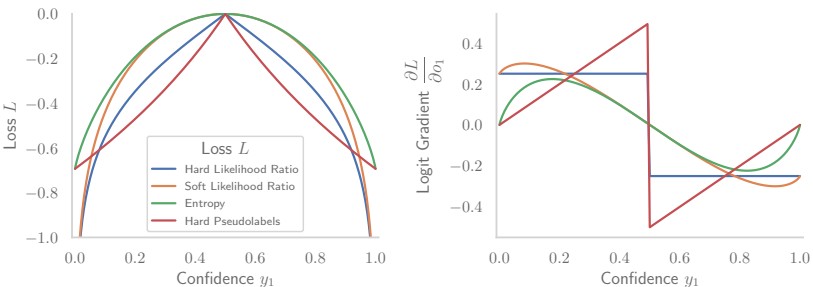

Figure 1: *Illustration of different losses for confidence maximization.* Losses (left, shifted such that maxima of all losses are at 0) and the resulting gradients with respect to the first logit (right) as a function of the first classes confidence are shown for the case of a binary classification problem. Both *entropy* and *hard pseudo-labels* have vanishing gradients for high confidence predictions[*]. Accordingly, both have maximum gradient amplitude for low-confidence self-supervision, with this effect being stronger for the hard pseudo-labels. *Hard Likelihood Ratio* has constant gradient amplitude for any confidence and thus takes into account low- and high-confidence self-supervision equally. *Soft Likelihood Ratio* also shows non-vanishing (albeit non-maximum) gradients for high-confidence self-supervision and additionally produces small gradient amplitudes from low-confidence self-supervision. Since the likelihood ratio-based losses are unbounded, the design of the model needs to ensure that logits cannot grow unbounded. [*]ResNet50 has a median prediction confidence of 0.929 on the ImageNet validation data and hence we consider a confidence of $\leq 0.8$ as *low* and $\geq 0.95$ as *high*.

respect to the logits $|\frac{\partial L_{pl}}{\partial o_{c^*}}|$ being generated on data where the network has low confidence (see Figure 1). This is undesirable since it corresponds to the network being affected most by data points where the network's self-supervision is least reliable[1].

An alternative is to use soft pseudo-labels, that is $y^r = \hat{y}$. This takes uncertainty in network predictions into account during self-labelling and results in the *entropy minimization* loss of TENT (Wang et al., 2020): $L_{ent}(\hat{y}) = H(\hat{y}, \hat{y}) = H(\hat{y}) = -\sum_c \hat{y}_c \log \hat{y}_c$. However, also for the entropy the logits' gradient magnitude $|\frac{\partial L_{ent}}{\partial o}|$ goes to 0 when one of the entries in $\hat{y}$ goes to 1 (see Figure 1). For a binary classification task, for instance, the maximal logits' gradient amplitude is obtained for $\hat{y} \approx (0.82, 0.18)$. This implies that during later stages of test-time adaptation where many predictions typically already have high confidence (significantly above 0.82), gradients are dominated by datapoints with relative low confidence in self-supervision.

While both hard and soft pseudo-labels are clearly motivated, they are not optimal in conjunction with a gradient-based optimizer since the self-supervision from low confidence predictions dominates (at least during later stages of training). We address this issue by proposing two losses that increase the gradient amplitude from high confidence predictions. We argue that this leads to stronger self-supervision (better gradient direction when averaged over the batch) than from the entropy loss (see also Sec. A.1 for an illustrative example supporting this claim) . The two losses are analogous to $L_{pl}$ and $L_{ent}$, but are not based on the cross-entropy $H$ but on the negative log likelihood ratios:

$$R(\hat{y}, y^r) = -\sum_c y_c^r \log \frac{\hat{y}_c}{\sum_{i \neq c} \hat{y}_i} \quad = -\sum_c y_c^r (\log \hat{y}_c - \log \sum_{i \neq c} \hat{y}_i) = H(\hat{y}, y^r) + \sum_c y_c^r \log \sum_{i \neq c} \hat{y}_i$$

Note that while the entropy $H$ is lower bounded by 0, $R$ can get arbitrary small if $y_c^r \to 1$ and the sum $\sum_{i \neq c} \hat{y}_i \to 0$ and thus $\log \sum_{i \neq c} \hat{y}_i \to -\infty$. This property will induce non-vanishing gradients for high confidence predictions.

The first loss we consider is the *hard likelihood ratio* loss that is defined similarly to the hard pseudo-labels loss $L_{pl}$:

$$L_{hlr}(\hat{y}) = R(\hat{y}, \text{onehot}(\hat{y})) = -\log(\frac{\hat{y}_{c^*}}{\sum_{i \neq c^*} \hat{y}_i}) \quad = -\log(\frac{e^{o_{c^*}}}{\sum_{i \neq c^*} e^{o_i}}) = -o_{c^*} + \log \sum_{i \neq c^*} e^{o_i},$$

---

[1]The prediction confidence for a datapoint can be interpreted as a proxy for its distance to the decision boundary. A low confidence prediction indicates that a datapoint appears to be close to the decision boundary and the model is less certain on which side of the decision boundary the datapoint should lie. We call this "low confidence self-supervision" since the direction of the gradient becomes ambiguous.

where $c^* = \arg\max \hat{y}$. We note that $\frac{\partial L_{hlr}}{\partial o_{c^*}} = -1$, thus also high-confidence self-supervision contributes equally to the maximum logits' gradients. This loss was also independently proposed as negative log likelihood ratio loss by Yao et al. (2020) as a replacement to the fully-supervised cross entropy loss for classification task. However, to the best of our knowledge, we are the first to motivate and identify the advantages of this loss for self-supervised learning and test-time adaptation due to its non-saturating gradient property.

In addition to $L_{hlr}$, we also account for uncertainty in network predictions during self-labelling in a similar way as for the entropy loss $L_{ent}$, and propose the *soft likelihood ratio* loss:

$$L_{slr}(\hat{y}) = R(\hat{y}, \hat{y}) = -\sum_c \hat{y}_c \cdot \log(\frac{\hat{y}_c}{\sum_{i \neq c} \hat{y}_i}) \qquad = \sum_c \hat{y}_c(-o_c + \log \sum_{i \neq c} e^{o_i})$$

We note that as $\hat{y}_{c^*} \to 1$, $L_{slr}(\hat{y}) \to L_{hlr}(\hat{y})$. Thus the asymptotic behavior of the two likelihood ratio losses for high confidence predictions is the same. However, the soft likelihood ratio loss creates lower amplitude gradients for low confidence self-supervision. We provide illustrations of the discussed losses and the resulting logits' gradients in Figure 1. Furthermore, an illustration of other losses like the max square loss and Charbonnier penalty can be found in Sec. A.7.

We note that both likelihood ratio losses would typically encourage the network to simply scale its logits larger and larger, since this would reduce the loss even if the ratios between the logits remain constant. However, when finetuning an existing network and restricting the layers that are adapted such that the logits remain approximately scale-normalized, these losses can provide a useful and non-vanishing gradient signal for network adaptation. We achieve this appproximate scale normalization by freezing the top layers of the respective networks. In this case, normalization layers such as batch normalization prohibit "logit explosion". However, predicted confidences can presumably become overconfident; calibrating confidences in a self-supervised test-time adaptation setting is an open and important direction for future work.

## 4 EXPERIMENTAL SETTINGS

**Datasets** We evaluate our method on image classification datasets for corruption robustness and domain adaptation. We evaluate on the challenging benchmark ImageNet-C (Hendrycks & Dietterich, 2019), which includes a wide variety of 15 different synthetic corruptions with 5 severity levels that attribute to data shift. This benchmark also includes 4 additional corruptions as validation data. For domain adaptation, we choose ImageNet trained models to adapt to ImageNet-R proposed by Hendrycks et al. (2020). ImageNet-R comprises 30,000 image renditions for 200 ImageNet classes. Domain adaptation on VisDA-C (Peng et al., 2017) and digit classification can be found in Sec. A.6.

**Models** Our method operates in a fully test-time adaptation setting that allows us to use any arbitrary pretrained model. We use publicly available ImageNet pretrained models ResNet50, DenseNet121, ResNeXt50, MobileNetV2 from torchvision Torch-Contributors (2020). We also test on a robust ResNet50 model trained using DeepAugment+AugMix [2] Hendrycks et al. (2020).

**Baseline for fully test-time adaptation** Since TENT from Wang et al. (2020) outperformed competing methods and fits the fully test-time adaptation setting, we consider it as a baseline and compare our results to this approach. Similar to TENT, we also adapt model features by estimating the normalization statistics and optimize only the channel-wise affine parameters on the target distribution.

**Settings** We conduct test-time adaptation on a target distribution with both online and offline updates using the Adam optimizer with learning rate 0.0006 with batch size 64. We set the weight of $L_{conf}$ in our loss function to $\delta = 0.025$ and $\kappa = 0.9$ in the running estimate $p_t(y)$ of $L_{div}$ (we investigate the effect of $\kappa$ in the Sec. A.4). Similar to SHOT (Liang et al., 2020), we also choose the target distribution $p_{\mathcal{D}'}(y)$ in $L_{div}$ as a uniform distribution over the available classes. For TENT, we use SGD with momentum 0.9 at learning rate 0.00025 with batch size 64. These values correspond to the ones of Wang et al. (2020); alternative settings for TENT did not improve performance. For offline updates, we adapt the models for 5 epochs using a cosine decay schedule of the learning rate. We found that the models converge during 3 to 5 epochs and do not improve further. Similar to Wang et al. (2020), we also control for ordering by data shuffling and sharing the order across the methods.

---

[2]From https://github.com/hendrycks/imagenet-r. Owner permitted to use it for research/commercial purposes.

Table 1: Test-time adaptation of ResNet50 on ImageNet-C at highest severity level 5. Ground truth labels are used to adapt the model in supervised manner to obtain empirical upper bound performance.

| Method | Gauss | Shot | Impulse | Defocus | Glass | Motion | Zoom | Snow | Frost | Fog | Bright | Contrast | Elastic | Pixel | JPEG | mean |
|---|---|---|---|---|---|---|---|---|---|---|---|---|---|---|---|---|
| No Adaptation | 2.44 | 2.99 | 1.96 | 17.92 | 9.82 | 14.78 | 22.50 | 16.89 | 23.31 | 24.43 | 58.93 | 5.43 | 16.95 | 20.61 | 31.65 | 18.04 |
| Pseudo Labels | 2.44 | 2.99 | 1.96 | 17.92 | 9.82 | 14.78 | 22.50 | 16.89 | 23.31 | 24.43 | 58.93 | 5.43 | 16.95 | 20.61 | 31.65 | 18.04 |
| | | | | | Online adaptation (evaluation on a batch directly after adaptation on the batch) | | | | | | | | | | | |
| TENT | 28.60 | 31.06 | 30.54 | 29.09 | 28.07 | 42.32 | 50.39 | 48.01 | 42.05 | 58.40 | 68.20 | 27.25 | 55.68 | 59.46 | 53.64 | 43.51 |
| TENT+ | 29.09 | 31.65 | 30.68 | 29.33 | 28.65 | 42.32 | 50.32 | 48.09 | 42.54 | 58.39 | 68.23 | 31.43 | 55.90 | 59.46 | 53.68 | 43.98 |
| HLR (ours) | 33.10 | 36.08 | 34.74 | 33.21 | 33.31 | 46.36 | 52.77 | 51.42 | 45.47 | 60.01 | 68.07 | 42.75 | 58.02 | 60.42 | 55.34 | 47.40 |
| SLR (ours) | **35.11** | **37.93** | **36.83** | **35.13** | **35.13** | **48.29** | **53.45** | **52.68** | **46.52** | **60.74** | **68.40** | **44.78** | **58.74** | **61.13** | **55.97** | **48.72** |
| | | | | | Evaluation after epoch 1 | | | | | | | | | | | |
| TENT | 32.44 | 35.01 | 34.77 | 32.40 | 31.62 | 47.23 | 53.09 | 51.61 | 43.26 | 60.42 | 68.85 | 24.39 | 58.53 | 61.62 | 56.00 | 46.08 |
| TENT+ | 33.75 | 36.38 | 35.67 | 33.43 | 33.25 | 47.66 | 53.20 | 52.06 | 44.85 | 60.60 | **68.93** | 33.43 | 58.94 | 61.75 | 56.21 | 47.34 |
| HLR (ours) | 38.39 | 41.11 | 40.28 | 38.25 | 38.18 | 51.63 | 55.55 | 55.45 | 48.96 | 62.19 | 68.17 | 49.47 | 60.34 | 62.51 | 57.42 | 51.19 |
| SLR (ours) | **39.51** | **42.09** | **41.58** | **39.35** | **39.02** | **52.67** | **55.80** | **55.92** | **49.64** | **62.62** | 68.47 | **50.27** | **60.80** | **63.01** | **57.80** | **51.90** |
| | | | | | Evaluation after epoch 5 | | | | | | | | | | | |
| TENT | 30.64 | 33.80 | 34.72 | 30.13 | 29.05 | 49.08 | 53.63 | 52.86 | 38.47 | 61.13 | 68.81 | 10.72 | 59.25 | 62.15 | 56.44 | 44.72 |
| TENT+ | 35.19 | 38.12 | 37.43 | 34.82 | 34.95 | 50.33 | 54.24 | 53.88 | 46.28 | 61.50 | **69.07** | 29.87 | 60.01 | 62.61 | 57.09 | 48.35 |
| HLR (ours) | 41.37 | **44.04** | 43.68 | **41.74** | **41.09** | 54.26 | 56.43 | 57.03 | 50.81 | 63.05 | 68.29 | **50.98** | 61.15 | 63.08 | 58.13 | 53.0 |
| SLR (ours) | **41.52** | 42.90 | **44.07** | 41.69 | 40.78 | **54.76** | **56.59** | **57.35** | **51.01** | **63.53** | 68.72 | 50.65 | **61.49** | **63.46** | **58.32** | **53.12** |
| Groundtruth | 55.68 | 58.10 | 61.27 | 55.84 | 55.08 | 65.83 | 67.22 | 67.56 | 62.60 | 72.49 | 76.97 | 65.04 | 70.86 | 72.51 | 68.56 | 65.04 |

Note that all the hyperparameters are tuned solely on the validation corruptions of ImageNet-C that are disjoint from the test corruptions. As discussed in Section 3.2.2, we freeze all trainable parameters in the top layers of the networks to prohibit "logit explosion". Normalization statistics are still updated in these layers. Sec. A.3 provides more details regarding frozen layers in different networks.

Furthermore, we prepend a trainable input transformation module $d$ (cf. Sec. 3.1) to the network to partially counteract the data-shift. Note that the parameters of this module discussed in Sec. 3.1 are trainable and subject to optimization. This module is initialized to operate as an identity function prior to adaptation on a target distribution by choosing $\tau = 1$, $\gamma = 1$, and $\beta = 0$. We adapt the parameters of this module along with the channel-wise affine transformations and normalization statistics in an end-to-end fashion, solely using our proposed loss function along with the optimization details mentioned above. The architecture of this module is discussed in Sec. A.2.

Since $L_{\text{div}}$ is independent of $L_{\text{conf}}$, we also propose to combine $L_{\text{div}}$ with TENT, i. e. $L = L_{\text{div}} + L_{ent}$. We denote this as TENT+ and also set $\kappa = 0.9$ here. Note that TENT optimizes all channel-wise affine parameters in the network (since entropy is saturating and does not cause logit explosion). For a fair comparison to our method, we also freeze the top layers of the networks in TENT+. We show that adding $L_{\text{div}}$ and freezing top layers significantly improves the networks performance over TENT. Note that SHOT (Liang et al., 2020) is the combination of TENT, batch-level diversity regularizer, and their pseudo labeling strategy. TENT+ can be seen as a variant of SHOT but without the pseudo labeling. Please refer to Sec. A.5 for the test-time adaptation of pretrained models with SHOT.

Note that each corruption and severity in ImageNet-C is treated as a different target distribution and we reset model parameters to their pretrained values before every adaptation. We run our experiments for three times with random seeds (2020, 2021, 2022) in PyTorch and report the average accuracies.

## 5 RESULTS

**Evaluation on ImageNet-C** We adapt different models on the ImageNet-C benchmark using TENT, TENT+, and both *hard likelihood ratio* (HLR) and *soft likelihood ratio* (SLR) losses in an online adaptation setting. Figure 2 (top row) depicts the mean corruption accuracy (mCA%) of each model computed across all the corruptions and severity levels. It can be observed that TENT+ improves over TENT, showcasing the importance of a diversity regularizer $L_{\text{div}}$. Importantly, our methods HLR and SLR outperform TENT and TENT+ across DenseNet121, MobileNetV2, ResNet50, ResNeXt50 and perform comparable with TENT+ on robust ResNet50-DeepAugment+Augmix model. This shows that the mCA% of robust DeepAugment+Augmix model can be further increased from 58% (before adaptation) to 67.5% using test-time adaptation techniques. Here, the average of mCA obtained from three different random seeds are depicted along with the error bars. These smaller error bars represent that the test-time adaptation results are not sensitive to the choice of random seed.

We also illustrate the performance of ResNet50 on the highest severity level across all 15 test corruptions of ImageNet-C in Table 1. Here, online adaptation results along with the offline adaptation on epoch 1 and 5 are reported. It can be seen that online adaptation and single epoch of test-time

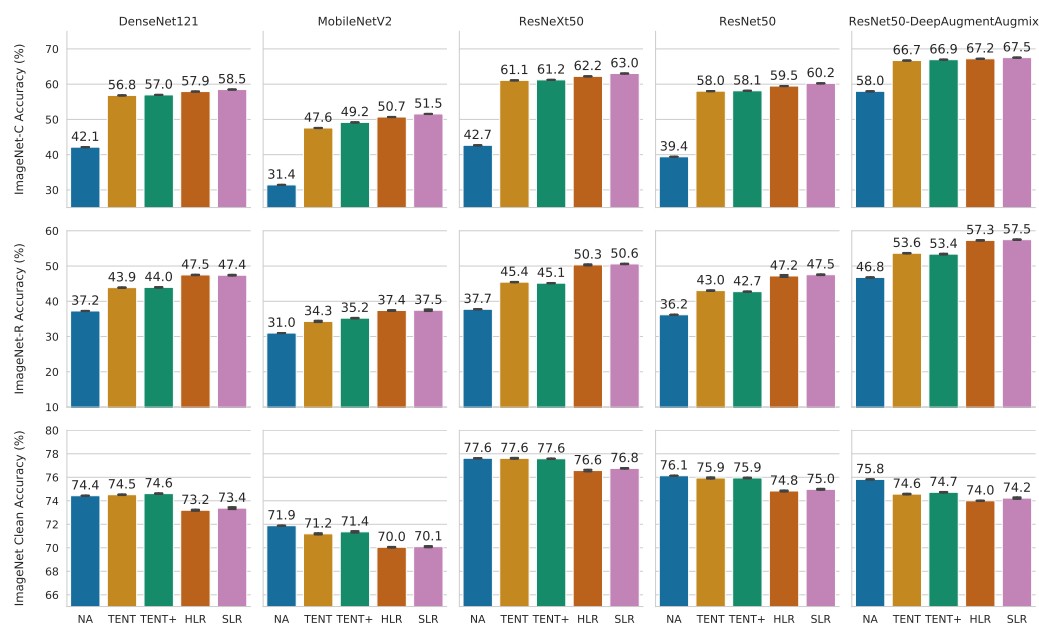

Figure 2: Test-time adaptation results on (top row) ImageNet-C, averaged across all 15 corruptions and severities, (middle row) ImageNet-R, (bottom row) clean ImageNet. NA refers to "No Adaptation".

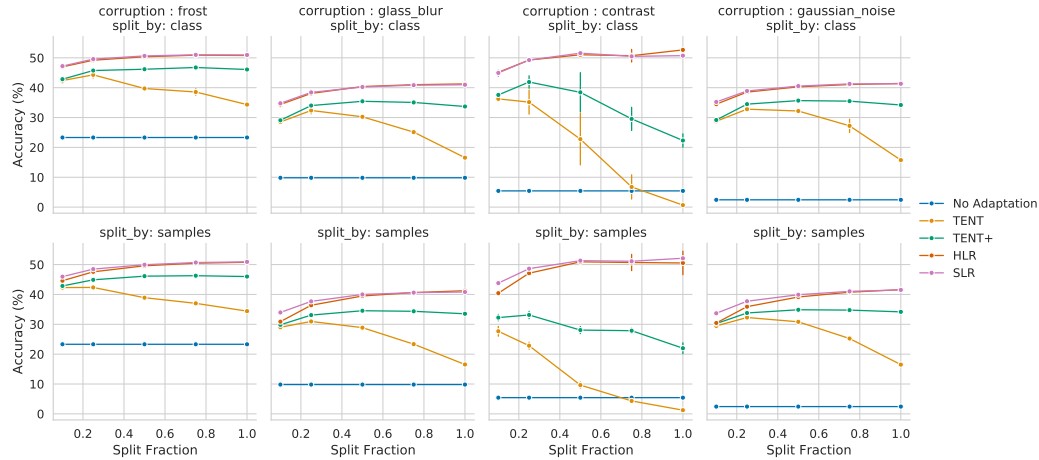

Figure 3: Test-time adaptation of ResNet50 using (top row) a subset of classes, and (bottom row) a subset of samples per class on 4 different corruptions at severity 5. Accuracy is computed based on the evaluation of adapted model on the entire target data. Note that error bars are smaller to visualize.

adaptation improves the performance significantly and makes minor improvements until epoch 5. TENT adaptation for more than one epoch result in reduced performance and TENT with $L_{\text{div}}$ (TENT+) prevents this behavior. Both HLR and SLR clearly and consistently outperform TENT / TENT+ on the ResNet50 and also note that SLR outweighs HLR. We also compare our results with the hard pseudo-labels (PL) objective and also with an oracle setting where the groundtruth labels of the target data are used for adapting the model in a supervised manner (GT). Note that this oracle setting is not of practical importance but illustrates the empirical upper bound on fully test-time adaptation performance under the chosen modulation parametrization.

**ImageNet-R** We online adapt different models on ImageNet-R and depict the results in Figure 2 (middle row). Results show that HLR and SLR clearly outperform TENT and TENT+ and significantly improve performance of all the models, including the model pretrained with DeepAugment+Augmix.

**Evaluation with data subsets** Above we evaluate the model on the same data that is also used for the test-time adaptation. Here, we test model generalization by adapting on a subset of target data

Table 2: SSIM and SLR-adapted ResNet50 accuracy without and with input transformation (IT).

| Corruption | Gauss | Shot | Impulse | Defocus | Glass | Motion | Zoom | Snow | Frost | Fog | Bright | Contrast | Elastic | Pixel | JPEG |
|---|---|---|---|---|---|---|---|---|---|---|---|---|---|---|---|
| SSIM | 0.123 | 0.147 | 0.135 | **0.623** | **0.648** | **0.622** | **0.676** | 0.517 | 0.575 | 0.619 | 0.653 | 0.545 | **0.625** | **0.786** | **0.800** |
| SSIM+IT | **0.173** | **0.188** | **0.347** | 0.605 | 0.638 | 0.603 | 0.670 | **0.580** | **0.628** | **0.626** | **0.676** | **0.765** | 0.616 | 0.776 | 0.795 |
| SLR | 41.52 | 42.90 | 44.07 | 41.69 | 40.78 | 54.76 | 56.59 | 57.35 | 51.01 | 63.53 | 68.72 | 50.65 | **61.49** | 63.46 | **58.32** |
| SLR+IT | **43.09** | **44.39** | **64.05** | **41.98** | **40.99** | **55.73** | **56.75** | **58.56** | **51.68** | **63.64** | **68.85** | **55.01** | 61.32 | **63.59** | 58.24 |

and evaluate the performance on the whole dataset (in offline setting), which also includes unseen data that is not used for adaptation. We conduct two case studies: (i) adapt on the data from a subset of ImageNet classes and evaluate the performance on the data from all the classes. (ii) Adapt only on a subset of data from each class and test on all seen and unseen samples from the whole dataset.

Figure 3 illustrates generalization of a ResNet50 adapted on different proportions of the data across different corruptions, both in terms of classes and samples. We observe that adapting a model on a small subset of samples and classes is sufficient to achieve reasonable accuracy on the whole target data. This suggests that the adaptation actually learns to compensate the data shift rather than overfitting to the adapted samples or classes. The performance of TENT decreases as the number of classes/samples increases, because $L_{ent}$ can converge to trivial collapsed solutions and more data corresponds to more updates steps during adaptation. Adding $L_{\text{div}}$ such as in TENT+ stabilizes the adaptation process and reduces this issues. Reported are the average of random seeds with error bars.

**Input transformation** We investigate whether the input transformation (IT) module, trained end-to-end with a ResNet50 and SLR loss on data of the respective distortion *without* seeing any source (undistorted) data, can partially undo certain domain shifts of ImageNet-C and also increase accuracy on corrupted data. We measure domain shift via the structural similarity index measure (SSIM) (Wang et al., 2004) between the clean image (unseen by the model) and its distorted version/the output of IT on the distorted version. Following offline adaptation setting, Table 2 shows that IT increases the SSIM considerably on certain distortions such as Impulse, Contrast, Snow, and Frost. IT increases SSIM also for other types of noise distortions, while it slightly reduces SSIM for the blur distortions, Elastic, Pixelate, and JPEG. When combined with SLR, IT considerably increases accuracy on distortions for which also SSIM increased significantly (for instance +20 percent points on Impulse, +4 percent points on Contrast) and never reduces accuracy by more than 0.11 percent points. More results on online and offline adaptation with TENT / TENT+ can be found in Table A3.

**Clean images** As a sanity check, we investigate the effect of test-time adaptation when target data comes from the same distribution as training data. For this, we online adapt pretrained models on clean validation data of ImageNet. The results in Figure 2 (bottom row) depict that the performance of SLR/HLR adapted models drops by 0.8 to 1.8 percent points compared to the pretrained model. We attribute this drop to self-supervision being less reliable than the original full supervision on in-distribution training data. The drop is smaller for TENT and TENT+, presumably because predictions on in-distribution target data are typically highly confident such that there is little gradient and thus little change to the pretrained networks by TENT. In summary, while self-supervision by confidence maximization is a powerful method for adaptation to domain shift, the observed drop when adapting to data from the source domain indicates that there is "no free lunch" in test-time adaptation.

## 6  CONCLUSION

We propose a method to improve corruption robustness and domain adaptation of models in a fully test-time adaptation setting. Unlike entropy minimization, our proposed loss functions provide non-vanishing gradients for high confident predictions and thus attribute to improved adaptation in a self-supervised manner. We also show that additional diversity regularization on the model predictions is crucial to prevent trivial solutions and stabilize the adaptation process. Lastly, we introduce a trainable input transformation module that partially refines the corrupted samples to support the adaptation. We show that our method improves corruption robustness on ImageNet-C and domain adaptation to ImageNet-R on different ImageNet models. We also show that adaptation on a small fraction of data and classes is sufficient to generalize to unseen target data and classes.

## 7 ETHICS STATEMENT

We abide by the general ethical principles listed by ICLR code of ethics. Our work does not include the study of human subjects, dataset releases, do not raise potential conflicts of interest, or discrimination/bias/fairness concerns, or privacy and security issues. Our non-saturating loss increases accuracy but might result in over confident predictions, which can cause harm in safety-critical downstream applications when not properly calibrated. At the same time, self-supervised confidence maximization might amplify bias in pretrained models. We hope that the diversity regularizer in the loss partially compensates this issue.

## 8 REPRODUCIBILITY STATEMENT

We provide complete details of our experimental setup for reproducibility. Sec. 4 provides details of the network architectures, optimizer, learning rate, batch size, choice of hyperparameters of our method and the random seeds used for generating the results. Sec. A.3 provides more details regarding frozen layers in different networks. Sec. A.2 shows the structure of input transformation module used in this work. We will also provide a link to an anonymous downloadable source code as a comment directed to the reviewers and area chairs in the discussion forum.

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

## A   APPENDIX

### A.1   ILLUSTRATIVE EXAMPLE OF LOG LIKELIHOOD RATIO ADAPTATION OBJECTIVE

A simple 1D example is devised to illustrate the benefits of proposed log likelihood ratio as test time adaptation objective. Consider data points (unlabeled) that are sampled from the following bimodal distribution: $0.5 \cdot \mathcal{N}(-1, 3) + 0.5 \cdot \mathcal{N}(+1, 3)$, that is: half of the samples come from a normal distribution with mean -1 and the other half from a normal distribution with mean +1 (and both having standard deviation 3). We can interpret these two components of the mixture distributions as corresponding to data of two different classes, but class labels are of course unavailable during unsupervised test-time adaptation.

We assume a simple logistic model of the form $p_\theta(y = 1|x) = \frac{1}{1+e^{-(x+\theta)}}$, where $x$ is the value of the data sample and $\theta$ is a scalar offset that determines the decision boundary. By construction, we know that the minimum density of the mixture distribution on $[-1, 1]$ is at 0. Since confidence maximization aims as moving the decision boundary to regions in input space with minimum data density (in this case to 0), we can compare different self-supervised confidence maximization losses in the finite data regime as follows: for every finite data sample with $N$ data points $\{x_i\}$ for $i = 1, \dots, N$ and loss function $L$, we solve $\theta^*(L) = \arg\min_{\theta \in [-1,1]} L(\theta, \{x_i\})$, where the loss (such as entropy or SLR) is averaged over all data points. The absolute value $|\theta^*(L)|$ gives us then an estimate of the error of the decision boundary parameter $|\theta^*(L)|$ for the given data set and loss function.

Table A1 provides this error for different loss functions and different number of data samples. It can be seen that SLR and HLR clearly outperform Entropy loss (TENT) for all data regimes. The difference between SLR and HLR is generally very small. While SLR seems to be consistently slightly better than HLR, this difference is not statistically significant. We attribute the superiority of SLR/HLR compared to entropy to the fact that all data points have non-saturating loss, regardless of their distance to the decision boundary. Thus, all data contributes to localizing the decision boundary, while for saturating losses such as the entropy, effectively only "nearby" points determine the decision boundary. This example illustrates that our proposed non-saturating losses are beneficial over entropy loss for self-supervised confidence maximization.

Table A1: Illustrates the error of the decision boundary parameter for different loss functions and different number of samples averaged over 100 runs (shown are mean and standard error of mean).

| #samples | 100 | 200 | 500 | 1000 | 2000 | 10000 | 20000 |
|---|---|---|---|---|---|---|---|
| Entropy | 0.487±0.031 | 0.364±0.029 | 0.230±0.018 | 0.152±0.013 | 0.117±0.009 | 0.052±0.004 | 0.033±0.003 |
| **HLR** | **0.357±0.023** | **0.234±0.018** | **0.145±0.012** | **0.094±0.008** | **0.071±0.006** | **0.032±0.002** | **0.022±0.002** |
| SLR | **0.332±0.022** | **0.214±0.017** | **0.140±0.011** | **0.088±0.008** | **0.067±0.006** | **0.032±0.002** | **0.021±0.002** |

### A.2   INPUT TRANSFORMATION MODULE

Note that we define our adaptable model as $g = f \circ d$, where $d$ is a trainable network prepended to a pretrained neural network $f$ (e.g., pretrained ResNet50). We choose $d(x) = \gamma \cdot [\tau x + (1 - \tau)r_\psi(x)] + \beta$, where $\tau \in \mathbb{R}$, $(\beta, \gamma) \in \mathbb{R}^{n_{in}}$ with $n_{in}$ being the number of input channels, $r_\psi$ being a network with identical input and output shape, and $\cdot$ denoting elementwise multiplication. Here, $\beta$ and $\gamma$ implement a channel-wise affine transformation and $\tau$ implements a convex combination of unchanged input and the transformed input $r_\psi(x)$. We set $\tau = 1$, $\gamma = \mathbf{1}$, and $\beta = \mathbf{0}$, to ensure that $d(x) = x$ and thus $g = f$ at initialization. In principle, $r_\psi$ can be chosen arbitrarily. Here, we choose $r_\psi$ as a simple stack of $3 \times 3$ convolutions with stride 1 and padding 1, group normalization, and ReLUs without any upsampling/downsampling layers. Specifically, the structure of $g$ is illustrated in Figure A1.

In addition to the results reported in Table 2, we also compare TENT and TENT+ with and without Input Transformation (IT) module on ResNet50 for all corruptions at severity level 5 in both online adaptation setting and offline adaptation with 5 epochs in Table A3. Furthermore, we also present the qualitative results of the image transformations from the input transformation module adapted with SLR (offline setting) in Figure A2.

Table A2: Ablation study on the components of input transformation module on ResNet50 for all corruptions at severity level 5.

| Corruption | Gauss | Shot | Impulse | Defocus | Glass | Motion | Zoom | Snow | Frost | Fog | Bright | Contrast | Elastic | Pixel | JPEG | mean |
|---|---|---|---|---|---|---|---|---|---|---|---|---|---|---|---|---|
| $x$ | 41.52 | 42.90 | 44.07 | 41.69 | 40.78 | 54.76 | 56.59 | 57.35 | 51.01 | 63.53 | 68.72 | 50.65 | 61.49 | 63.46 | 58.32 | 53.12 |
| $r_\psi(x)$ | 13.17 | 26.57 | 28.81 | 5.09 | 3.61 | 30.61 | 49.79 | 53.73 | 45.96 | 58.82 | 65.79 | 53.73 | 56.77 | 60.14 | 53.38 | 40.40 |
| $\tau x + (1-\tau)r_\psi(x)$ | 43.13 | 46.43 | 56.25 | 41.80 | 40.90 | 55.75 | 56.65 | 58.55 | 51.72 | 63.59 | 68.83 | 53.89 | 61.50 | 63.73 | 58.51 | 54.74 |
| $\gamma \cdot [\tau x + (1-\tau)r_\psi(x)] + \beta$ | 43.18 | 46.24 | 56.21 | 41.91 | 40.89 | 55.79 | 56.66 | 58.50 | 51.72 | 63.56 | 68.83 | 54.26 | 61.49 | 63.76 | 58.52 | 54.76 |

Table A3: Test-time adaptation of ResNet50 on ImageNet-C at highest severity level 5 with and without Input Transformation (IT) module. Reported are the mean accuracy(%) across three random seeds (2020/2021/2022). While IT also improves performance when combined with TENT+, it is still clearly outperformed by SLR+IT.

| Method | Gauss | Shot | Impulse | Defocus | Glass | Motion | Zoom | Snow | Frost | Fog | Bright | Contrast | Elastic | Pixel | JPEG |
|---|---|---|---|---|---|---|---|---|---|---|---|---|---|---|---|
| Online adaptation (evaluation on a batch directly after adaptation on the batch) | | | | | | | | | | | | | | | |
| TENT | 28.60 | 31.06 | 30.54 | 29.09 | 28.07 | 42.32 | 50.39 | 48.01 | 42.05 | 58.40 | 68.20 | 27.25 | 55.68 | 59.46 | 53.64 |
| TENT + IT | 28.99 | 31.73 | 31.15 | 28.87 | 27.85 | 42.43 | 50.36 | 48.02 | 41.95 | 58.37 | 68.19 | 24.35 | 55.68 | 59.49 | 53.57 |
| TENT+ | 29.09 | 31.65 | 30.68 | 29.33 | 28.65 | 42.32 | 50.32 | 48.09 | 42.54 | 58.39 | 68.23 | 31.43 | 55.90 | 59.46 | 53.68 |
| TENT+ + IT | 29.48 | 32.34 | 31.38 | 29.06 | 28.42 | 42.43 | 50.33 | 48.11 | 42.47 | 58.40 | 68.20 | 32.11 | 55.87 | 59.49 | 53.64 |
| SLR (ours) | 35.11 | 37.93 | 36.83 | 35.13 | **35.13** | 48.29 | 53.45 | 52.68 | 46.52 | **60.74** | **68.40** | 44.78 | 58.74 | 61.13 | **55.97** |
| SLR + IT (ours) | **36.19** | **39.17** | **40.46** | **35.17** | 34.87 | **48.67** | **53.62** | **52.71** | 46.93 | 60.66 | 68.30 | **46.55** | **58.79** | **61.27** | 55.93 |
| Evaluation after epoch 5 | | | | | | | | | | | | | | | |
| TENT | 30.64 | 33.80 | 34.72 | 30.13 | 29.05 | 49.08 | 53.63 | 52.86 | 38.47 | 61.13 | 68.81 | 10.72 | 59.25 | 62.15 | 56.44 |
| TENT + IT | 31.92 | 36.02 | 38.14 | 30.44 | 28.68 | 49.04 | 53.59 | 52.99 | 38.76 | 61.14 | 68.84 | 13.52 | 59.23 | 62.15 | 56.56 |
| TENT+ | 35.19 | 38.12 | 37.43 | 34.82 | 34.95 | 50.33 | 54.24 | 53.88 | 46.28 | 61.50 | **69.07** | 29.87 | 60.01 | 62.61 | 57.09 |
| TENT+ + IT | 36.13 | 39.84 | 41.03 | 34.62 | 34.72 | 50.33 | 54.10 | 53.91 | 46.46 | 61.54 | **69.07** | 30.22 | 59.95 | 62.72 | 57.11 |
| SLR (ours) | 41.52 | 42.90 | 44.07 | 41.69 | 40.78 | 54.76 | 56.59 | 57.35 | 51.01 | 63.53 | 68.72 | 50.65 | **61.49** | 63.46 | **58.32** |
| SLR+IT (ours) | **43.09** | **44.39** | **64.05** | **41.98** | **40.99** | 55.73 | 56.75 | 58.56 | 51.68 | 63.64 | 68.85 | **55.01** | 61.32 | **63.59** | 58.24 |

### A.2.1 CONTRIBUTION OF EACH COMPONENT IN INPUT TRANSFORMATION MODULE

Table A2 shows the results of ablation study on the components of input transformation module on ResNet50 for all corruptions at severity level 5 adapted with SLR for 5 epochs. The ablation study includes: (1) no input transformation module $d(x) = x$, (2) with network $d(x) = r_\psi(x)$, (3) including $\tau$, (4) including channel-wise affine transformation $\gamma$ and $\beta$. We can observe that the inputs transformed with network $r_\psi$ drops the performance without the convex combination with $\tau$. The additional channel wise affine transformations didn't bring further consistent improvements and can be ignored from the transformation module. Exploring other architectural choices and training (or pretraining) strategy for the input transformation module would be an interesting avenue for future work.

## A.3 FROZEN LAYERS IN DIFFERENT NETWORKS

As discussed in Section 3.2.2, we freeze all trainable parameters in the top layers of the networks to prohibit "logit explosion". That implies, we do not optimize the channel-wise affine transformations of the top layers but normalization statistics are still estimated. Similar to the hyperparameters of test time adaptation settings, the choice of these layers are made using ImageNet-C validation data. We mention the frozen layers of each architecuture below. Note that the naming convention of these layers are based on the model definition in torchvision:

- DenseNet121 - *features.denseblock4, features.norm5*.
- MobileNetV2 - *features.16, features.17, features.18*.
- ResNeXt50, ResNet50 and ResNet50 (DeepAugment+Augmix) - *layer4*.

### A.3.1 RESULTS WITHOUT FREEZING THE TOP LAYERS

We mentioned that the proposed losses could alternatively encourage the network to scale the logits grow larger and larger and still reduce the loss. However, we did not find any considerable differences empirically in the explored settings when adapting the model with or without freezing the top layer. We found that adapting the model with and without freezing the top layers have comparable performance in both online and offline adaptation settings as shown in Table A4 respectively. However, we would still recommend freezing the top-most layers as the default choice to be on the safe side. These results indicate that the early layers capture the distribution shift sufficiently to improve the model adaptation.

Table A4: Comparing the online and offline adaptation results with and without freezing the affine parameters of top normalization layers of ResNet50 at severity 5. Here, "Freeze" and "NoFreeze" refer to the setting with and without freezing the top affine layers respectively.

| Corruption | Gauss | Shot | Impulse | Defocus | Glass | Motion | Zoom | Snow | Frost | Fog | Bright | Contrast | Elastic | Pixel | JPEG | mean |
|---|---|---|---|---|---|---|---|---|---|---|---|---|---|---|---|---|
| | | | | | | Online evaluation | | | | | | | | | | |
| TENT+ NoFreeze | 29.05 | 31.32 | 30.32 | 28.95 | 28.29 | 42.37 | 50.45 | 48.12 | 42.21 | 58.51 | 68.29 | 28.17 | 55.57 | 59.47 | 53.46 | 43.63 |
| TENT+ Freeze | 29.21 | 31.54 | 30.55 | 29.17 | 28.60 | 42.54 | 50.47 | 48.18 | 42.51 | 58.50 | 68.30 | 31.25 | 55.76 | 59.54 | 53.62 | 43.98 |
| HLR NoFreeze | 33.73 | 36.50 | 35.63 | 33.99 | 33.88 | 46.55 | 52.76 | 51.44 | 45.82 | 59.74 | 67.37 | 43.19 | 57.69 | 59.77 | 54.95 | 47.53 |
| HLR Freeze | 33.10 | 36.08 | 34.74 | 33.21 | 33.31 | 46.36 | 52.77 | 51.42 | 45.47 | 60.01 | 68.07 | 42.75 | 58.02 | 60.42 | 55.34 | 47.40 |
| SLR NoFreeze | 35.61 | 38.37 | 37.50 | 35.83 | 35.81 | 48.29 | 53.61 | 52.62 | 46.85 | 60.42 | 67.71 | 44.93 | 58.43 | 60.56 | 55.65 | 48.81 |
| SLR Freeze | 35.11 | 37.93 | 36.83 | 35.13 | 35.13 | 48.29 | 53.45 | 52.68 | 46.52 | 60.74 | 68.40 | 44.78 | 58.74 | 61.13 | 55.97 | 48.72 |
| | | | | | | offline evaluation | | | | | | | | | | |
| TENT+ NoFreeze | 32.03 | 35.33 | 35.28 | 31.92 | 31.27 | 49.20 | 53.79 | 53.01 | 40.37 | 61.22 | 68.79 | 19.38 | 59.25 | 62.20 | 56.51 | 45.97 |
| TENT+ Freeze | 35.19 | 38.12 | 37.43 | 34.82 | 34.95 | 50.33 | 54.24 | 53.88 | 46.28 | 61.50 | 69.07 | 29.87 | 60.01 | 62.61 | 57.09 | 48.35 |
| HLR NoFreeze | 41.60 | 43.80 | 43.89 | 42.21 | 41.50 | 53.82 | 56.21 | 56.71 | 50.83 | 62.74 | 67.87 | 51.34 | 60.65 | 62.58 | 57.70 | 52.89 |
| HLR Freeze | 41.37 | 44.04 | 43.68 | 41.74 | 41.09 | 54.26 | 56.43 | 57.03 | 50.81 | 63.05 | 68.29 | 50.98 | 61.15 | 63.08 | 58.13 | 53.0 |
| SLR NoFreeze | 41.45 | 43.95 | 44.26 | 42.56 | 41.60 | 54.25 | 56.13 | 56.72 | 50.92 | 62.97 | 68.02 | 50.99 | 60.90 | 62.83 | 57.86 | 53.02 |
| SLR Freeze | 41.52 | 42.90 | 44.07 | 41.69 | 40.78 | 54.76 | 56.59 | 57.35 | 51.01 | 63.53 | 68.72 | 50.65 | 61.49 | 63.46 | 58.32 | 53.12 |

## A.4 EFFECT OF $\kappa$

Note that the running estimate of $L_{\mathrm{div}}$ prevents model collapsed to trivial solutions i.e., model predicts only a single or a set of classes as outputs regardless of the input samples. $L_{\mathrm{div}}$ encourages model to match it's empirical distribution of predictions to class distribution of target data (uniform distribution in our experiments). Such diversity regularization is crucial as there is no direct supervision attributing to different classes and thus aids to avoid collapsed trivial solutions. In Figure A3, we investigate different values of $\kappa$ on validation corruptions of ImageNet-C to study its effectiveness on our approach. It can be observed that both the HLR and SLR without $L_{\mathrm{div}}$ leads to collapsed solutions (e.g., accuracy drops to $0\%$) on some of the corruptions and the performance gains are not consistent across all the corruptions. On the other hand, $L_{\mathrm{div}}$ with $\kappa = 0.9$ remain consistent and improve the performance across all the corruptions.

## A.5 TEST-TIME ADPTATION OF PRETRAINED MODELS WITH SHOT

Following SHOT (Liang et al., 2020), we use their pseudo labeling strategy on the ImageNet pretrained ResNet50 in combination with TENT+, HLR and SLR. Note that TENT+ and pseudo labeling strategy jointly forms the method SHOT. The pseudo labeling strategy starts after the 1st epoch and thereafter computed at every epoch. The weight for the loss computed on the pseudo labels is set to 0.3, similar to (Liang et al., 2020). Different values for this weight is explored and found 0.3 to perform best. Table A6 compares the results of the methods with and without pseudo labeling strategy. It can be observed that the results with pseudo labeling strategy perform worse than without taking this strategy into account.

We further modified the pretrained ResNet50 by following the network modifications suggested in (Liang et al., 2020), that includes adding a bottleneck layer with BatchNorm and applying weight norm on the linear classifier along with smooth label training to facilitate the pseudo labeling strategy. Table A7 shows that the pseudo labeling strategy on such network improve the results of TENT+ from epoch 1 to epoch 5. However, there are no improvements noticed in SLR. Moreover, Table A8 shows that NO pseudo labeling strategy on the same network performs better than applying the pseudo labeling strategy. Finally, the no pseduo labeling results from Table A6 and A8 shows that additional modifications to ResNet50 do not improve the performance when compared to the standard ResNet50.

## A.6 DOMAIN ADAPTATION ON VISDA-C AND DIGIT CLASSIFICATION

**VisDA-C:** We extended our experiments to VisDA-C. We followed similar network architecture from SHOT (Liang et al., 2020) and evaluated TENT+, our SLR loss function with diversity regularizer. Similar to ImageNet-C, we adapted only the channel wise affine parameters of batchnorm layers for 5 epochs with Adam optimizer with cosine decay scheduler of the learning rate with initial value $2e-5$. Here, the batchsize is set to 64, the weight of $L_{\mathrm{conf}}$ in our loss function to $\delta = 0.25$ and $\kappa = 0$ in the running estimate $p_t(y)$ of $L_{\mathrm{div}}$, since the number of classes in this dataset (12 classes) is smaller than the batchsize. Setting $\kappa = 0$ enables the batch wise diversity regularizer. Table A9 shows

Table A5: Test-time adaptation of ResNet50 on ImageNet-C at highest severity level 5. Same as Table 1 with error bars.

| name | | | Epoch 1 | | | | Epoch 5 | | | |
|---|---|---|---|---|---|---|---|---|---|---|
| corruption | No adaptation | PL | TENT | TENT+ | HLR | SLR | TENT | TENT+ | HLR | SLR |
| Gauss | 2.44 | 2.44 | 32.44±0.10 | 33.75±0.09 | 38.39±0.25 | **39.51**±0.23 | 30.64±0.51 | 35.19±0.17 | 41.37±0.09 | **41.52**±0.08 |
| Shot | 2.99 | 2.99 | 35.01±0.17 | 36.38±0.19 | 41.11±0.13 | **42.09**±0.26 | 33.80±0.74 | 38.12±0.10 | **44.04**±0.09 | 42.90±0.08 |
| Impulse | 1.96 | 1.96 | 34.77±0.09 | 35.67±0.15 | 40.28±0.20 | **41.58**±0.04 | 34.72±1.01 | 37.43±0.09 | 43.68±0.06 | **44.07**±0.06 |
| Defocus | 17.92 | 17.92 | 32.40±0.10 | 33.43±0.14 | 38.25±0.32 | **39.35**±0.13 | 30.13±0.61 | 34.82±0.25 | **41.74**±0.12 | 41.69±0.07 |
| Glass | 9.82 | 9.82 | 31.62±0.15 | 33.25±0.01 | 38.18±0.08 | **39.02**±0.09 | 29.05±0.21 | 34.95±0.13 | **41.09**±0.17 | 40.78±0.08 |
| Motion | 14.78 | 14.78 | 47.23±0.11 | 47.66±0.12 | 51.63±0.08 | **52.67**±0.25 | 49.08±0.08 | 50.33±0.07 | 54.26±0.02 | **54.76**±0.04 |
| Zoom | 22.50 | 22.50 | 53.09±0.06 | 53.20±0.07 | 55.55±0.06 | **55.80**±0.07 | 53.63±0.16 | 54.24±0.06 | 56.43±0.07 | **56.59**±0.05 |
| Snow | 16.89 | 16.89 | 51.61±0.05 | 52.06±0.09 | 55.45±0.11 | **55.92**±0.06 | 52.86±0.13 | 53.88±0.07 | 57.03±0.12 | **57.35**±0.03 |
| Frost | 23.31 | 23.31 | 43.26±0.30 | 44.85±0.20 | 48.96±0.07 | **49.64**±0.14 | 38.47±0.50 | 46.28±0.27 | 50.81±0.08 | **51.01**±0.02 |
| Fog | 24.43 | 24.43 | 60.42±0.08 | 60.60±0.05 | 62.19±0.03 | **62.62**±0.04 | 61.13±0.08 | 61.50±0.05 | 63.05±0.04 | **63.53**±0.08 |
| Bright | 58.93 | 58.93 | 68.85±0.02 | **68.93**±0.03 | 68.17±0.01 | 68.47±0.05 | 68.81±0.06 | **69.07**±0.06 | 68.29±0.09 | 68.72±0.10 |
| Contrast | 5.43 | 5.43 | 24.39±0.98 | 33.43±0.77 | 49.47±0.20 | **50.27**±0.08 | 10.72±0.32 | 29.87±1.36 | **50.98**±2.54 | 50.65±0.55 |
| Elastic | 16.95 | 16.95 | 58.53±0.05 | 58.94±0.05 | 60.34±0.18 | **60.80**±0.08 | 59.25±0.06 | 60.01±0.02 | 61.15±0.04 | **61.49**±0.07 |
| Pixel | 20.61 | 20.61 | 61.62±0.06 | 61.75±0.07 | 62.51±0.10 | **63.01**±0.08 | 62.15±0.04 | 62.61±0.08 | 63.08±0.06 | **63.46**±0.08 |
| JPEG | 31.65 | 31.65 | 56.00±0.09 | 56.21±0.05 | 57.42±0.13 | **57.80**±0.04 | 56.44±0.07 | 57.09±0.02 | 58.13±0.09 | **58.32**±0.05 |

Table A6: Test-time adaptation of ResNet50 on ImageNet-C at highest severity level 5 with and without the pseudo labeling strategy (Liang et al., 2020).

| name | | No pseudo labeling: Epoch 5 | | | Pseudo labeling: Epoch 5 | | |
|---|---|---|---|---|---|---|---|
| corruption | No adaptation | TENT+ | HLR | SLR | TENT+ | HLR | SLR |
| Gauss | 2.44 | 33.97±0.17 | 41.37±0.09 | 41.52±0.08 | 34.08±0.11 | 34.88±0.35 | 35.58±0.06 |
| Shot | 2.99 | 37.95±0.10 | 44.04±0.09 | 42.90±0.08 | 36.74±0.26 | 37.61±0.49 | 37.98±0.19 |
| Impulse | 1.96 | 36.93±0.09 | 43.68±0.06 | 44.07±0.06 | 36.69±0.04 | 37.24±0.22 | 37.77±0.05 |
| Defocus | 17.92 | 32.69±0.25 | 41.74±0.12 | 41.69±0.07 | 33.99±0.28 | 34.76±0.11 | 35.11±0.10 |
| Glass | 9.82 | 33.36±0.13 | 41.09±0.17 | 40.78±0.08 | 34.06±0.12 | 34.51±0.30 | 34.81±0.27 |
| Motion | 14.78 | 51.42±0.07 | 54.26±0.02 | 54.76±0.04 | 50.91±0.09 | 48.96±0.39 | 49.46±0.20 |
| Zoom | 22.50 | 54.33±0.06 | 56.43±0.07 | 56.59±0.05 | 54.10±0.10 | 52.49±0.02 | 52.50±0.23 |
| Snow | 16.89 | 54.55±0.07 | 57.03±0.12 | 57.35±0.03 | 54.06±0.08 | 52.49±0.19 | 52.95±0.07 |
| Frost | 23.31 | 45.80±0.27 | 50.81±0.08 | 51.01±0.02 | 44.44±0.07 | 45.47±0.26 | 46.06±0.20 |
| Fog | 24.43 | 62.09±0.05 | 63.05±0.04 | 63.53±0.08 | 61.91±0.08 | 59.66±0.14 | 59.98±0.12 |
| Bright | 58.93 | 69.03±0.06 | 68.29±0.09 | 68.72±0.10 | 68.98±0.02 | 65.59±0.06 | 66.00±0.03 |
| Contrast | 5.43 | 24.08±1.36 | 50.98±2.54 | 50.65±0.55 | 29.37±0.95 | 44.58±0.38 | 45.64±0.47 |
| Elastic | 16.95 | 60.36±0.02 | 61.15±0.04 | 61.49±0.07 | 60.23±0.05 | 57.48±0.14 | 57.87±0.04 |
| Pixel | 20.61 | 63.10±0.08 | 63.08±0.06 | 63.46±0.08 | 62.98±0.04 | 59.72±0.02 | 60.05±0.14 |
| JPEG | 31.65 | 57.21±0.02 | 58.13±0.09 | 58.32±0.05 | 57.09±0.04 | 54.72±0.09 | 54.88±0.07 |

average results from three different random seeds and also shows that SLR outperforms TENT+ on this dataset.

**Domain adaptation from SVHN to MNIST / MNIST-M / USPS:** ResNet26 is trained on SVHN dataset for 50 epochs with batch size 128, SGD optimizer with momentum 0.9 and initial learning rate 0.01, which drops to 0.001 and 0.0001 at 25th and 40th epoch respectively. ResNet26 obtains 96.49% test accuracy on SVHN. Domain adaptation of SVHN trained ResNet26 to MNIST/MNIST-M/USPS

Table A7: Test-time adaptation of modified ResNet50 (following (Liang et al., 2020)) on ImageNet-C at highest severity level 5 with pseudo labeling strategy at epoch 1 and epoch 5.

| name | | Pseudo labeling: Epoch 1 | | | Pseudo labeling: Epoch 5 | | |
|---|---|---|---|---|---|---|---|
| corruption | No adaptation | TENT+ | HLR | SLR | TENT+ | HLR | SLR |
| Gauss | 2.95 | 31.03±0.18 | 34.65±0.28 | 37.21±0.23 | 35.26±0.16 | 35.93±0.23 | 37.61±0.30 |
| Shot | 3.65 | 33.55±0.07 | 38.09±0.30 | 40.30±0.09 | 37.39±0.05 | 38.95±0.16 | 40.42±0.06 |
| Impulse | 2.54 | 32.70±0.07 | 36.95±0.05 | 39.73±0.07 | 38.16±0.08 | 38.13±0.04 | 40.12±0.11 |
| Defocus | 19.36 | 31.66±0.15 | 35.08±0.05 | 37.18±0.15 | 35.95±0.17 | 36.72±0.13 | 37.96±0.25 |
| Glass | 9.72 | 31.06±0.06 | 35.46±0.12 | 37.62±0.10 | 35.98±0.04 | 36.84±0.11 | 37.90±0.02 |
| Motion | 15.66 | 46.96±0.12 | 49.95±0.12 | 51.87±0.14 | 52.24±0.02 | 51.90±0.12 | 52.76±0.09 |
| Zoom | 22.20 | 52.45±0.02 | 54.15±0.22 | 54.84±0.18 | 54.80±0.07 | 54.84±0.09 | 54.95±0.14 |
| Snow | 17.56 | 51.79±0.05 | 53.98±0.06 | 55.44±0.04 | 55.15±0.02 | 55.27±0.20 | 55.75±0.02 |
| Frost | 24.11 | 45.59±0.06 | 47.87±0.03 | 48.96±0.11 | 48.10±0.20 | 48.52±0.11 | 49.13±0.20 |
| Fog | 25.59 | 60.33±0.03 | 61.55±0.10 | 62.21±0.16 | 62.39±0.03 | 62.38±0.12 | 62.38±0.11 |
| Bright | 58.30 | 68.84±0.04 | 68.44±0.04 | 68.60±0.10 | 69.13±0.04 | 68.50±0.02 | 68.47±0.09 |
| Contrast | 6.49 | 42.34±0.19 | 47.98±0.13 | 50.32±0.28 | 42.11±0.15 | 49.22±0.42 | 50.80±0.19 |
| Elastic | 17.72 | 58.47±0.02 | 59.70±0.06 | 60.30±0.03 | 60.40±0.04 | 60.27±0.22 | 60.45±0.21 |
| Pixel | 21.29 | 61.39±0.06 | 62.10±0.07 | 62.71±0.10 | 63.04±0.02 | 62.71±0.07 | 62.81±0.07 |
| JPEG | 32.13 | 55.22±0.03 | 56.49±0.07 | 57.04±0.07 | 57.21±0.06 | 57.25±0.07 | 57.37±0.05 |

Table A8: Test-time adaptation of modified ResNet50 (following (Liang et al., 2020)) on ImageNet-C at highest severity level 5 with and without pseudo labeling strategy.

| name | | No Pseudo labeling: Epoch 5 | | | Pseudo labeling: Epoch 5 | | |
|---|---|---|---|---|---|---|---|
| corruption | No adaptation | TENT+ | HLR | SLR | TENT+ | HLR | SLR |
| Gauss | 2.95 | 34.96±0.08 | 38.58±0.12 | 39.72±0.13 | 35.26±0.16 | 35.93±0.23 | 37.61±0.30 |
| Shot | 3.65 | 37.22±0.17 | 41.59±0.09 | 42.45±0.05 | 37.39±0.05 | 38.95±0.16 | 40.42±0.06 |
| Impulse | 2.54 | 37.82±0.04 | 40.88±0.07 | 42.39±0.03 | 38.16±0.08 | 38.13±0.04 | 40.12±0.11 |
| Defocus | 19.36 | 34.46±0.12 | 39.22±0.15 | 39.78±0.09 | 35.95±0.17 | 36.72±0.13 | 37.96±0.25 |
| Glass | 9.72 | 35.12±0.05 | 38.83±0.13 | 39.37±0.07 | 35.98±0.04 | 36.84±0.11 | 37.90±0.02 |
| Motion | 15.66 | 51.91±0.09 | 53.23±0.05 | 54.00 | 52.24±0.02 | 51.90±0.12 | 52.76±0.09 |
| Zoom | 22.20 | 54.57±0.05 | 55.76±0.04 | 55.79±0.02 | 54.80±0.07 | 54.84±0.09 | 54.95±0.14 |
| Snow | 17.56 | 55.02±0.05 | 56.35±0.12 | 56.80±0.04 | 55.15±0.02 | 55.27±0.20 | 55.75±0.02 |
| Frost | 24.11 | 48.18±0.09 | 49.86±0.22 | 50.43±0.08 | 48.10±0.20 | 48.52±0.11 | 49.13±0.20 |
| Fog | 25.59 | 62.24±0.04 | 62.90±0.06 | 63.29±0.06 | 62.39±0.03 | 62.38±0.12 | 62.38±0.11 |
| Bright | 58.30 | 69.12±0.01 | 68.72±0.06 | 68.83±0.05 | 69.13±0.04 | 68.50±0.02 | 68.47±0.09 |
| Contrast | 6.49 | 33.91±0.92 | 52.13±0.16 | 53.04±0.14 | 42.11±0.15 | 49.22±0.42 | 50.80±0.19 |
| Elastic | 17.72 | 60.37±0.11 | 60.89±0.08 | 61.12±0.01 | 60.40±0.04 | 60.27±0.22 | 60.45±0.21 |
| Pixel | 21.29 | 62.97±0.02 | 62.95±0.05 | 63.21±0.05 | 63.04±0.02 | 62.71±0.07 | 62.81±0.07 |
| JPEG | 32.13 | 57.10±0.06 | 57.91±0.06 | 57.99±0.11 | 57.21±0.06 | 57.25±0.07 | 57.37±0.05 |

Table A9: Performance on VisDA-C dataset

| Method | Accuracy(%) |
|---|---|
| No Adaptation | 46.1 |
| TENT+ | 81.83±0.16 |
| SLR | 82.32±0.16 |

is conducted with Adam optimizer with constant learning rate 0.001 for 20 epochs on TENT, TENT+ and SLR with three random seeds (2020/2021/2022). Table A10 compares our proposed loss SLR with TENT variants on ResNet26 and shows that our approach outperforms them across all the datasets.

## A.7 COMPARISON WITH MAX SQUARE LOSS AND CHARBONNIER PENALTY

Similar to Figure 1, we provide the illustration of max square loss (Chen et al., 2019) and charbonnier penalty applied to entropy minimization (Yang & Soatto, 2020) in Figure A4. Both the max Squares loss and the charbonnier penalty applied to entropy minimization have gradients of the loss w.r.t. the logit that go to 0 for high confidence predictions (confidences greater than 0.95). This is not the case for our proposed non-saturating losses. This vanishing gradient cause high confidence predictions to have less contribution on the test-time adaptation than lower confident predictions. We also compare the online test-time adaptation results of these losses with our proposed losses in Table A11.

## A.8 TESTS ON IMAGENET-A

We adapted ResNet50 on ImageNet-A dataset for 5 epochs with different losses as presented in the Table A12. We see that SLR outperforms the other losses. However, the improvements from the test-time adaptation on this dataset are minimal when compared to the results on corrupted datasets. Thus, test-time adaptation is best suited when there is a shift in appearance in the inputs rather involving high-level or semantic changes as in ImageNet-A.

Table A10: Digit domain adaptation from SVHN to MNIST / MNIST-M / USPS. Reported values are mean accuracy(%) over three random seeds (2020/2021/2022).

| | MNIST | MNIST-M | USPS |
|---|---|---|---|
| No adaptation | 42.48 | 47.43 | 11.83 |
| TENT | 93.5 | 56.9 | 84.0 |
| TENT+ | 96.9 | 67.4 | 85.6 |
| SLR (ours) | **98.3** | **77.4** | **94.2** |

Table A11: Test-time online adaptation of ResNet50 on ImageNet-C at highest severity level 5. Similar to TENT+, $L_{\text{div}}$ is combined with Max Square loss and Charbonnier Penalty and also freeze top layers of the network. We denote these settings as Max Square+ and Charbonnier Penalty+. Here, different $\eta \in \{0.1, 0.3, 0.75, 1.0, 1.75, 2.0\}$ values are explored for Charbonnier penalty and found $\eta = 0.3$ performs better. Reported values are mean accuracy over three random seeds (2020/2021/2022) of ResNet50 at severity level 5. Results show that SLR outperforms Max Squares loss and the Charbonnier Penalty, even when tuning the Charbonnier penalty's hyperparamter $\eta$ carefully.

| Corruption | Gauss | Shot | Impulse | Defocus | Glass | Motion | Zoom | Snow | Frost | Fog | Bright | Contrast | Elastic | Pixel | JPEG |
|---|---|---|---|---|---|---|---|---|---|---|---|---|---|---|---|
| Max square | 17.14 | 18.50 | 17.68 | 17.58 | 17.40 | 30.02 | 42.81 | 38.15 | 36.03 | 52.23 | 66.73 | 21.22 | 47.66 | 53.24 | 45.02 |
| Max square+ | 16.85 | 18.15 | 17.35 | 17.29 | 17.08 | 29.59 | 42.34 | 37.67 | 35.68 | 51.74 | 66.58 | 20.95 | 47.28 | 52.78 | 44.46 |
| Charbonnier Penalty (eta=0.3) | 24.82 | 27.00 | 26.03 | 25.34 | 24.48 | 38.36 | 48.47 | 45.40 | 40.76 | 56.98 | 67.90 | 28.92 | 53.49 | 57.96 | 51.38 |
| Charbonnier Penalty+ (eta=0.3) | 24.55 | 26.76 | 25.45 | 24.86 | 24.22 | 38.00 | 48.22 | 44.98 | 40.55 | 56.80 | 67.83 | 30.22 | 53.33 | 57.77 | 51.14 |
| TENT | 28.60 | 31.06 | 30.54 | 29.09 | 28.07 | 42.32 | 50.39 | 48.01 | 42.05 | 58.40 | 68.20 | 27.25 | 55.68 | 59.46 | 53.64 |
| TENT+ | 29.09 | 31.65 | 30.68 | 29.33 | 28.65 | 42.32 | 50.32 | 48.09 | 42.54 | 58.39 | 68.23 | 31.43 | 55.90 | 59.46 | 53.68 |
| HLR (ours) | 33.10 | 36.08 | 34.74 | 33.21 | 33.31 | 46.36 | 52.77 | 51.42 | 45.47 | 60.01 | 68.07 | 42.75 | 58.02 | 60.42 | 55.34 |
| SLR (ours) | **35.11** | **37.93** | **36.83** | **35.13** | **35.13** | **48.29** | **53.45** | **52.68** | **46.52** | **60.74** | **68.40** | **44.78** | **58.74** | **61.13** | **55.97** |

Table A12: ResNet50 adaptation on ImageNet-A (reported is the model accuracy)

| | No adaptation | TENT | TENT+ | HLR | SLR |
|---|---|---|---|---|---|
| ImageNet-A | 0.0% | 0.04% | 0.08% | 0.51% | 0.55% |

## A.9 ADDITIONAL STUDIES ON TENT/TENT+

**Optimizer and learning rate:** The default optimization setting proposed for TENT (Wang et al., 2020) is SGD with learning rate (lr) 0.00025. Table A13 provide additional results for TENT and TENT+ results with both SGD and Adam at different learning rates for both online and offline (5 epochs) adaptation settings on ResNet50 for all corruptions at severity level 5. We can notice that SGD shown to perform better than Adam for TENT and slightly better for TENT+. Among different learning rates with SGD, higher learning rates bring additional improvements for TENT and TENT+ in online adaptation setting but they are still outperformed by our SLR results. However, for offline updates, the higher learning rates shown to hurt the model performance with TENT / TENT+ and our SLR results are superior over them. Note that our optimizer and learning rate remain same for both online and offline adaptation settings. Here, TENT behaves differently in online and offline adaptation for the same learning rate and our HLR/SLR behaves the same in both the settings.

**Results without freezing top layers:** We run TENT by freezing all but the affine parameters of normalization layers and we freeze top affine layers of the network for TENT+ additionally. In Table A14, we present the results of TENT+ without freezing the top affine layers of ResNet50 for all corruptions at severity level 5 with different learning rates using SGD. We can observe that the network with freezing top layers has comparable or sometimes better performance than no freeze network. These results indicate that the distribution shift is better captured in early affine layers and handling the distribution shift in deeper affine layers with high learning rate limit their functionality to operate on abstract representations and thereby affect the model performance.

Table A13: Evaluation of TENT and TENT+ with both SGD and Adam at different learning rate on ResNet50 for all corruptions at severity level 5.

| Method | Gauss | Shot | Impulse | Defocus | Glass | Motion | Zoom | Snow | Frost | Fog | Bright | Contrast | Elastic | Pixel | JPEG | mean |
|---|---|---|---|---|---|---|---|---|---|---|---|---|---|---|---|---|
| online adaptation TENT - SGD |||||||||||||||||
| lr 0.0001 | 24.76 | 26.97 | 25.93 | 25.13 | 24.37 | 37.74 | 47.98 | 44.73 | 40.26 | 56.43 | 67.62 | **27.51** | 52.88 | 57.34 | 50.85 | 40.7 |
| lr 0.00025 | 28.70 | 30.97 | 30.38 | 29.03 | 28.00 | 42.46 | 50.42 | 48.15 | 42.03 | 58.48 | 68.25 | 26.97 | 55.66 | 59.52 | 53.54 | 43.5 |
| lr 0.0004 | 30.01 | 32.35 | 32.17 | 30.11 | **29.25** | 44.32 | 51.37 | 49.42 | **42.18** | 59.18 | 68.43 | 24.19 | 56.68 | 60.26 | 54.45 | **44.29** |
| lr 0.0006 | **30.25** | **32.90** | **32.84** | 30.12 | 29.19 | 45.52 | 51.94 | 50.21 | 41.05 | 59.55 | **68.52** | 19.11 | 57.24 | 60.62 | 54.92 | 44.26 |
| lr 0.0008 | 29.59 | 32.48 | 32.73 | 29.37 | 28.38 | 46.07 | **52.05** | 50.59 | 40.45 | 59.70 | 68.48 | 15.03 | 57.54 | 60.78 | **55.10** | 43.88 |
| lr 0.001 | 28.11 | 31.13 | 32.06 | 28.56 | 26.96 | **46.21** | 52.03 | **50.72** | 39.33 | **59.74** | 68.40 | 12.36 | **57.66** | **60.85** | **55.10** | 43.28 |
| online adaptation TENT - Adam |||||||||||||||||
| lr 0.0001 | 26.71 | 29.30 | 28.60 | 26.75 | 26.04 | 40.99 | 50.39 | 47.77 | 40.60 | 58.43 | 68.49 | 23.44 | 55.61 | 59.13 | 53.18 | 42.36 |
| lr 0.00025 | 25.06 | 28.47 | 29.17 | 25.50 | 23.63 | 43.54 | 51.32 | 49.40 | 37.51 | 59.39 | 68.27 | 14.83 | 56.91 | 60.05 | 54.37 | 41.82 |
| lr 0.0004 | 18.53 | 21.34 | 24.40 | 19.65 | 16.34 | 41.61 | 49.67 | 47.94 | 31.70 | 58.77 | 67.65 | 9.30 | 56.46 | 59.66 | 53.69 | 38.44 |
| lr 0.0006 | 12.41 | 14.23 | 15.49 | 13.28 | 10.68 | 34.22 | 44.63 | 41.62 | 21.65 | 56.65 | 66.20 | 6.31 | 52.08 | 57.12 | 50.41 | 33.13 |
| lr 0.0008 | 8.98 | 10.05 | 10.82 | 9.76 | 8.08 | 24.09 | 37.15 | 32.72 | 15.69 | 49.32 | 63.38 | 4.42 | 44.82 | 53.36 | 42.22 | 27.65 |
| lr 0.001 | 6.92 | 7.57 | 8.34 | 7.52 | 6.41 | 18.46 | 28.07 | 25.61 | 12.55 | 40.22 | 57.01 | 3.35 | 35.87 | 45.79 | 31.53 | 22.34 |
| online adaptation TENT+ - SGD |||||||||||||||||
| lr 0.0001 | 24.73 | 26.91 | 25.63 | 24.86 | 24.31 | 37.55 | 47.79 | 44.49 | 40.15 | 56.30 | 67.57 | 28.67 | 52.80 | 57.19 | 50.70 | 40.64 |
| lr 0.00025 | 29.21 | 31.54 | 30.55 | 29.17 | 28.60 | 42.54 | 50.47 | 48.18 | 42.51 | 58.50 | 68.30 | **31.25** | 55.76 | 59.54 | 53.62 | 43.98 |
| lr 0.0004 | 30.95 | 33.35 | 32.66 | 30.90 | 30.43 | 44.67 | 51.49 | 49.63 | 43.54 | 59.29 | 68.48 | 31.21 | 56.98 | 60.37 | 54.60 | 45.23 |
| lr 0.0006 | 32.03 | 34.66 | 33.91 | 31.82 | 31.57 | 46.10 | 52.18 | 50.69 | 44.20 | 59.79 | 68.62 | 30.01 | 57.69 | 60.92 | 55.31 | 45.96 |
| lr 0.0008 | 32.37 | 35.16 | 34.58 | **32.00** | 31.91 | 46.96 | 52.50 | 51.28 | 44.55 | 60.08 | **68.68** | 27.23 | 58.09 | 61.21 | 55.66 | **46.15** |
| lr 0.001 | **32.52** | **35.32** | **34.84** | 31.89 | **32.00** | 47.41 | **52.56** | 51.61 | **44.66** | 60.24 | 68.66 | 24.75 | **58.34** | **61.45** | **55.82** | 46.13 |
| lr 0.002 | 31.03 | 34.69 | 33.39 | 29.41 | 29.80 | **48.12** | 52.47 | **52.11** | 42.96 | 60.42 | 68.48 | 15.25 | 58.57 | 61.67 | 55.74 | 44.94 |
| online adaptation TENT+ - Adam |||||||||||||||||
| lr 0.0001 | 27.95 | 30.48 | 29.30 | 27.86 | 27.80 | 41.57 | 50.64 | 48.07 | 42.44 | 58.65 | 68.53 | 29.24 | 55.98 | 59.31 | 53.38 | 43.41 |
| lr 0.00025 | 31.07 | 33.60 | 33.10 | 30.19 | 30.52 | 45.54 | 52.36 | 50.63 | 44.12 | 60.00 | 68.65 | 28.67 | 58.02 | 60.78 | 55.13 | 45.49 |
| lr 0.0004 | 31.31 | 33.90 | 33.59 | 30.00 | 30.29 | 46.60 | 52.46 | 51.41 | 43.65 | 60.26 | 68.43 | 25.66 | 58.25 | 61.08 | 55.44 | 45.48 |
| lr 0.0006 | 30.27 | 33.14 | 32.65 | 28.15 | 28.47 | 46.85 | 51.80 | 51.48 | 42.39 | 60.09 | 68.04 | 18.16 | 58.11 | 61.02 | 55.17 | 44.38 |
| lr 0.0008 | 28.21 | 31.20 | 30.62 | 24.61 | 25.18 | 46.42 | 50.79 | 50.92 | 40.09 | 59.79 | 67.67 | 13.48 | 57.60 | 60.70 | 54.66 | 42.79 |
| lr 0.001 | 25.31 | 28.53 | 27.92 | 20.80 | 21.05 | 45.29 | 49.57 | 49.89 | 37.94 | 59.30 | 67.15 | 9.77 | 57.11 | 60.20 | 53.86 | 40.91 |
| online adaptation Ours - Adam lr 0.0006 |||||||||||||||||
| HLR | 33.10 | 36.08 | 34.74 | 33.21 | 33.31 | 46.36 | 52.77 | 51.42 | 45.47 | 60.01 | 68.07 | 42.75 | 58.02 | 60.42 | 55.34 | 47.40 |
| SLR | **35.11** | **37.93** | **36.83** | **35.13** | **35.13** | **48.29** | **53.45** | **52.68** | **46.52** | **60.74** | **68.40** | **44.78** | **58.74** | **61.13** | **55.97** | **48.72** |
| offline adaptation TENT+ - SGD |||||||||||||||||
| lr 0.001 | 27.81 | 33.79 | 32.94 | 25.14 | 26.52 | 50.95 | 52.69 | 54.19 | 43.29 | 62.07 | 68.78 | 2.66 | 60.24 | 63.06 | 56.94 | 44.07 |
| lr 0.0008 | 31.80 | 35.84 | 35.01 | 27.64 | 29.49 | 51.61 | 53.33 | 54.47 | 44.57 | 62.15 | 68.89 | 9.03 | 60.40 | 62.98 | 57.07 | 45.6 |
| lr 0.00025 | 35.19 | 38.12 | 37.43 | 34.82 | 34.95 | 50.33 | 54.24 | 53.88 | 46.28 | 61.50 | 69.07 | 29.87 | 60.01 | 62.61 | 57.09 | 48.35 |
| offline adaptation Ours - Adam lr 0.0006 |||||||||||||||||
| HLR | 41.37 | **44.04** | 43.68 | **41.74** | **41.09** | 54.26 | 56.43 | 57.03 | 50.81 | 63.05 | 68.29 | **50.98** | 61.15 | 63.08 | 58.13 | 53.0 |
| SLR | **41.52** | 42.90 | **44.07** | 41.69 | 40.78 | **54.76** | **56.59** | **57.35** | **51.01** | **63.53** | 68.72 | 50.65 | **61.49** | **63.46** | **58.32** | **53.12** |

Table A14: Evaluation of TENT+ with and without freezing the top affine layers at different learning rate on ResNet50 for all corruptions at severity level 5. Here, "Freeze" and "NoFreeze" refer to the setting with and without freezing the top affine layers respectively.

| Method | Gauss | Shot | Impulse | Defocus | Glass | Motion | Zoom | Snow | Frost | Fog | Bright | Contrast | Elastic | Pixel | JPEG | mean |
|---|---|---|---|---|---|---|---|---|---|---|---|---|---|---|---|---|
| online adaptation TENT+ - SGD lr 0.00025 |||||||||||||||||
| NoFreeze | 29.05 | 31.32 | 30.32 | 28.95 | 28.29 | 42.37 | 50.45 | 48.12 | 42.21 | 58.51 | 68.29 | 28.17 | 55.57 | 59.47 | 53.46 | 43.63 |
| Freeze | 29.21 | 31.54 | 30.55 | 29.17 | 28.60 | 42.54 | 50.47 | 48.18 | 42.51 | 58.50 | 68.30 | 31.25 | 55.76 | 59.54 | 53.62 | 43.98 |
| online adaptation TENT+ - SGD lr 0.0004 |||||||||||||||||
| NoFreeze | 30.38 | 32.86 | 32.17 | 30.26 | 29.64 | 44.44 | 51.42 | 49.45 | 43.01 | 59.12 | 68.48 | 27.83 | 56.75 | 60.32 | 54.42 | 44.70 |
| Freeze | 30.95 | 33.35 | 32.66 | 30.90 | 30.43 | 44.67 | 51.49 | 49.63 | 43.54 | 59.29 | 68.48 | 31.21 | 56.98 | 60.37 | 54.60 | 45.23 |
| online adaptation TENT+ - SGD lr 0.0006 |||||||||||||||||
| NoFreeze | 30.87 | 33.72 | 32.89 | 30.58 | 29.76 | 45.58 | 52.01 | 50.32 | 43.03 | 59.47 | 68.60 | 25.66 | 57.35 | 60.71 | 54.88 | 45.02 |
| Freeze | 32.03 | 34.66 | 33.91 | 31.82 | 31.57 | 46.10 | 52.18 | 50.69 | 44.20 | 59.79 | 68.62 | 30.01 | 57.69 | 60.92 | 55.31 | 45.96 |
| online adaptation TENT+ - SGD lr 0.0008 |||||||||||||||||
| NoFreeze | 30.85 | 33.58 | 32.98 | 29.87 | 29.50 | 46.28 | 52.18 | 50.69 | 42.66 | 59.66 | 68.52 | 22.56 | 57.64 | 60.85 | 55.08 | 44.86 |
| Freeze | 32.37 | 35.16 | 34.58 | 32.00 | 31.91 | 46.96 | 52.50 | 51.28 | 44.55 | 60.08 | 68.68 | 27.23 | 58.09 | 61.21 | 55.66 | 46.15 |
| online adaptation TENT+ - SGD lr 0.001 |||||||||||||||||
| NoFreeze | 30.48 | 33.07 | 32.58 | 29.42 | 28.66 | 46.54 | 52.09 | 50.77 | 41.84 | 59.74 | 68.41 | 18.57 | 57.72 | 60.97 | 55.13 | 44.39 |
| Freeze | 32.52 | 35.32 | 34.84 | 31.89 | 32.00 | 47.41 | 52.56 | 51.61 | 44.66 | 60.24 | 68.66 | 24.75 | 58.34 | 61.45 | 55.82 | 46.13 |

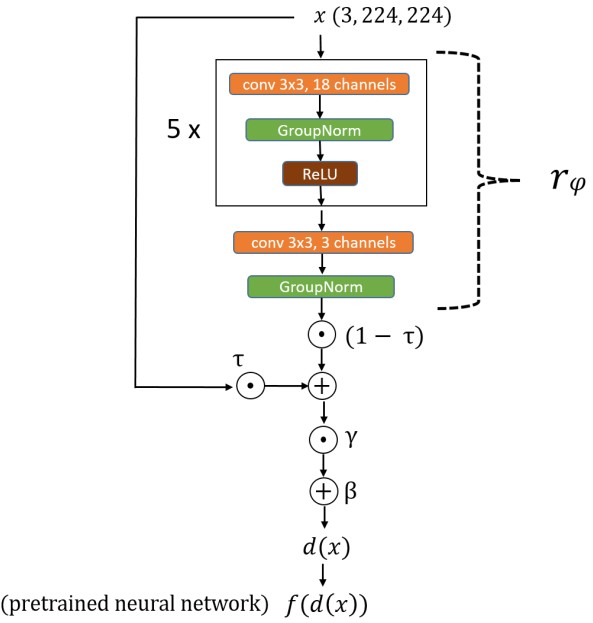

Figure A1: Structure of our adaptable model $g$, that comprises of $r_\psi$.

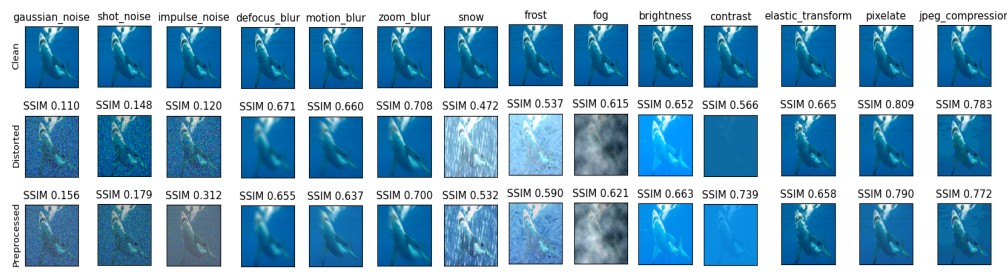

Figure A2: Qualitative results of image transformation from input transformation module adapted with SLR.

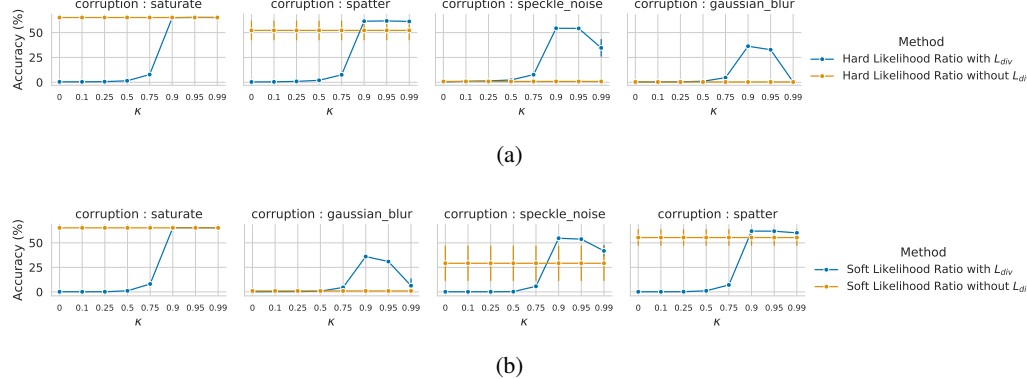

Figure A3: Effect of different $\kappa$ on both (a) HLR and (b) SLR

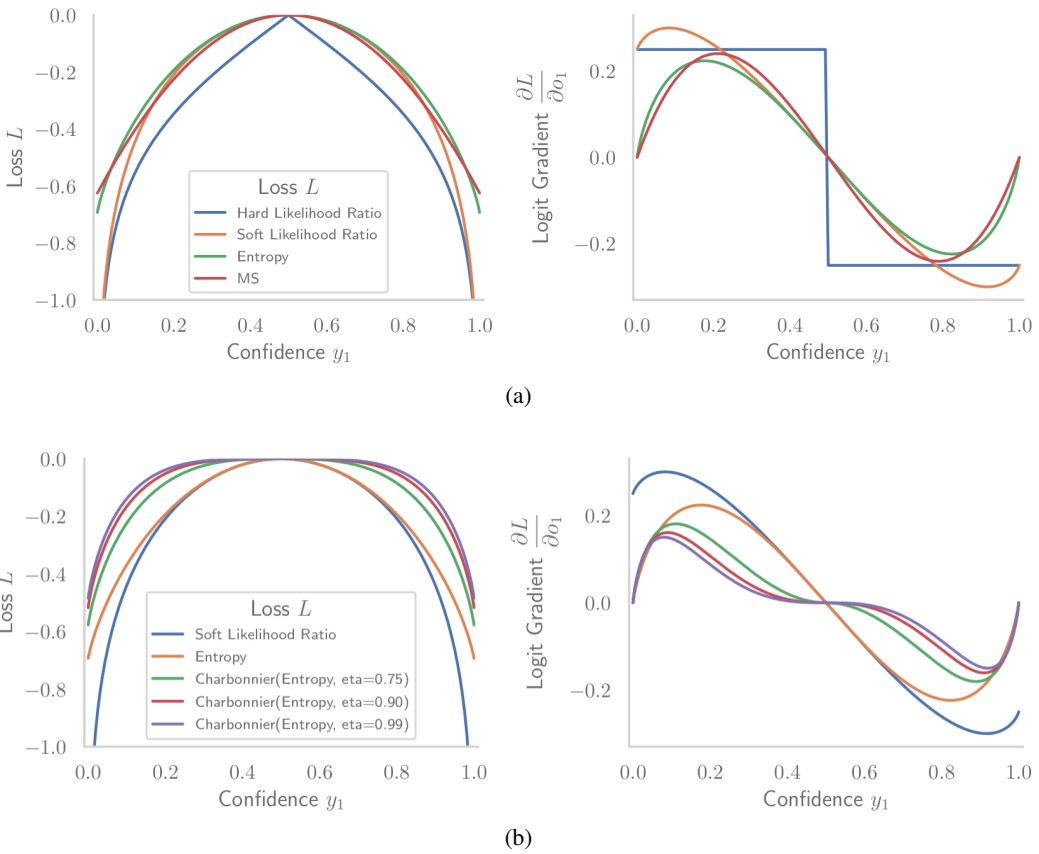

(a)

(b)

Figure A4: (a) Illustration of Max Square (MS) loss and (b) Charbonnier penalty with different $\eta$. Similar to entropy in Figure 1, both the losses have vanishing gradient for high confidence predictions (confidences greater than 0.95).

