# OpenReview forum: "Test-Time Adaptation to Distribution Shifts by Confidence Maximization and Input Transformation"
_ICLR.cc/2022/Conference — ICLR 2022 Submitted_

### Official Review · Reviewer_9MXJ · 2021-11-02

**Correctness:** 3
**Technical Novelty And Significance:** 2
**Empirical Novelty And Significance:** 2
**Recommendation:** 6
**Confidence:** 5

**Main Review:**

Strengths
+ The joint adaptation of input transformations and model parameters is novel for test-time adaptation. Neither TTT (Sun et al. 2020), test-time normalization (Scheider et al. 2020), nor TENT (Wang et al. 2021) adapt the input. The related work section covers prior projects that learn to transform the input during training rather than testing.
+ The proposed techniques—the alternative losses, the input transformation model, and diversity regularizer—improve accuracy jointly and separately. For instance, the input transformation model helps the proposed approach as well as TENT, and the diversity regularizer likewise fixes cases where TENT can fail. While these are not wholly new, it is still informative to double-check their effectiveness, ablate their combination, and show small but consistent improvement across multiple architectures and datasets (ImageNet-C, ImageNet-R, and the digit datasets MNIST/MNIST-M/USPS).
+ The proposed extensions are still online and efficient, so this method seemingly could be deployed as easily as TENT. The only counter to this is that small accuracy drops are reported on unshifted data, where these drops are not seen (TENT, BN) or smaller (TTT) for other test-time methods.
+ The results for test-time adaptation by optimization (TENT, HLR/SLR) on ImageNet-R are empirically new. Only test-time normalization (Schneider et al. 2020) had reported test-time results on this data.
+ Design choices are justified with experimental results (main tables), visualization (Figure 1 for losses and gradients), and toy experiments (Appendix A.1).

Weaknesses
- The novelty of the proposed extensions is diminished by intersection with prior methods. The losses for HLR and SLR are from Yao et al. 2020, although this work is the first to bring them to fully test-time adaptation, and that is worthwhile. (For comparison, note that TENT argued for entropy which is obviously well-established as a loss). The diversity regularizer is a core part of SHOT (Liang et al. 2020), although this work calculates it differently with a moving average. The input transformation model is the most new, as input adaptation of this kind has not been done during testing, but there are close connections to training like ANT (Rusak et al.) and CyCADA (Tzeng et al.).
- The proposed losses HLR and SLR further restrict the choice of parameters to adapt, as these losses would otherwise cause the predicted logits to grow without bound. Deriving an alternative loss without this restriction would be better to simplify the application of the method.
- The amount of improvement on shifted data is marginal, at 2-3 points absolute in many cases, while there is harm on the unshifted/standard/clean data. This argues against the motivation for non-saturating losses. This is admitted on pg. 9 under "Clean Images" but is not remedied. A new loss or combination of losses that improves shifted accuracy without hurting clean accuracy would be more significant.
- Input transformation (Sec. 3.1) is not closely studied or ablated. For instance, what if only the channel-wise input changes are included without the network? Do shallower networks do better or worse? Does input transformation help on all corruptions, or can it hurt?

For Rebuttal
- Please provide a control experiment for the choice of optimization. How do TENT and TENT+ fare when optimized with Adam, the same solver as HLR/SLR? For TENT/TENT+, does raising the learning rate for SGD or Adam improve results by counteracting the saturation of the entropy loss? The need for non-saturating losses is a key claim so this is worth double-checking.
- Please provide the results for TENT/TENT+ without freezing the weights of the deeper layers. The need for this freezing is a limitation of HLR and SLR, not TENT, so it is worth knowing if these additional parameters help the baselines.
- Please comment on the learned transformation models, and in particular the learned transformation weight tau. How much do the scale gamma and shift beta help on their own? As a more novel part of this work, the input transformation module deserves more analysis.
- Please explain the data subset experiments in more detail (Figure 3). Why does TENT fail as the split fraction reaches 1.0? In the TENT paper, there are generalization results with adaptation on target train and evaluation on target test, and the method still helps in that case. What is the justification for these use cases? Would it not be better to always adapt on all the data that is encountered?

Miscellaneous Feedback
- [clarity] please summarize results with the mean where appropriate, for instance by including the mean over corruption types in Table 1
- [clarity] consider including qualitative results of the learned image transformations, for instance in the appendix, to show the types and degrees of transformation.
- [text] in the related work, change "domain adaptation train" to "domain adaptation methods train"
- [text] in the related work, change "such setting refrain the cost" to "such settings spare the cost"
- [text] in Sec. 3.2.1, change "One option are" to "One option is"

**Summary Of The Paper:**

Test-time adaptation by entropy minimization can help models adapt to dataset shifts like corruptions without altering training. This work extends tent, an entropy minimization method, by proposing alternative non-saturating losses, adding a diversity regularizer, and adapting the input data along with the model parameters. The input is adapted by applying a convolutional image transformation model between the input and classification model. These extensions do not need more optimization iterations or supervision than the baselines: the method adapts online and efficiently without auxiliary supervision. Experiments on the corruption benchmark ImageNet-C and the newer benchmark ImageNet-R report reduced generalization error. The improvements are there but marginal, and they are consistent across multiple baseline architectures (ResNet, DenseNet, MobileNet, etc.). However, the clean accuracy reduced, so the proposed method does not strictly dominate prior work.

**Summary Of The Review:**

This work makes a reasonable but minor contribution that can inform further extensions of test-time adaptation. A large part of this work is the double-checking of elements of test-time adaptation methods, with only marginal empirical novelty and significance, or the importing of techniques from other scopes, with either no or only marginal technical novelty and significance. It is a pity that the most new part, the test-time input transformer, is not further studied and improved to give this work a more independent dimension of contribution. At the same time, the harm to accuracy on normal data is cause for hesitation to accept the proposed changes. While there is value in this work, rejection is recommended so that (1) the input transformer can be more fully covered and (2) the issue of improving out-of-distribution accuracy at the cost of in-distribution accuracy can be resolved.

**Final Review** The rebuttal thoroughly clarified results and offered additional experiments to empirically justify the contributions and reduce worries about potential issues. In particular, the results with the baselines of TENT/TENT+ now make sense with the clarification of online/offline results (please underline this in the paper), and the proposed method is not so sensitive to the choice of frozen layers. I have raised my score to 6, and I would have considered a 7 if there were such a rating. I did not go higher because the trade-off of lower accuracy on unshifted data for higher accuracy on shifted data remains. Nevertheless there is informative material here, in the main paper and appendix, and so I vote for acceptance so that this work can inform the burgeoning direction of test-time adaptation. Along with the more novel parts, such as input transformation during testing, this work also helpfully confirms and tunes other parts like the diversity regularizer in ways that future work can simply adopt.

---

> ### Author Response · Authors · 2021-11-22
> **Thank you for raising interesting questions. It further refines the understanding of our approach.**
>
> **1)Novelty of the proposed extensions:** We request the reviewer to refer the 1st response to Reviewer 8M4e.
>
> **2)TENT and TENT+ with SGD and Adam at different learning rates:**
> We provide TENT and TENT+ results in Section A.9 in appendix with both SGD and Adam at different learning rates. We notice that SGD shown to perform better than Adam for TENT and slightly better for TENT+. Among different learning rates with SGD, higher learning rates bring additional improvements in online adaptation setting but they are still outperformed by our SLR results. However, for offline updates, the higher learning rates shown to hurt the model performance with TENT / TENT+ and our SLR results are superior over them. Here, TENT behaves differently in online and offline adaptation for the same learning rate and our HLR/SLR behaves the same in both the settings.
>
> **3)Results for TENT+ without freezing the weights of the deeper layers:**
> We present the results of TENT+ without freezing the top affine layers of ResNet50 in Table A14 under Section A.9 in appendix. We can notice that the network with freezing top layers has comparable or sometimes better performance than no freeze network.
>
> **HLR and SLR further restrict the choice of parameters to adapt?**
> We mentioned in the paper that the proposed losses could alternatively encourage the network to scale the logits grow larger and larger and still reduce the loss. However, we did not find any considerable differences empirically in the explored settings when adapting the model with or without freezing the top layer. We found that adapting the model with and without freezing the top layers have comparable performance in both online and offline adaptation settings as shown in Table A4 under Section A.3.1 in appendix.  However, we would still recommend freezing the top-most layers as the default choice to be on the safe side.
>
> Note that we used publicly available pretrained models to adapt to distribution shifts and freezing the top-most layers did not require any architectural changes. The main change compared to TENT is that we restrict the choice of adaptable layers a bit further - TENT freezes all but the affine parameters of normalization layers; we freeze some of the top affine layers additionally. Moreover, we  show that the results of TENT+ , HLR and SLR are comparable with and without freezing the top layers of the network. These results indicate that the early layers capture the distribution shift sufficiently to improve the  model adaptation.
>
> **4) Analysis on the input transformation module:**
>    Thanks for suggesting us to perform the ablation study. We have noticed that the  additional channel wise affine transformations didn't bring further consistent improvements and can be ignored from the transformation module. We have included the results from this ablation study in our revised version in Appendix A.2.1 .
>
> **5)Why does TENT fail as the split fraction reaches 1.0 under data subset experiments?**
> Note that the data subset experiment is an offline evaluation setting  where the model is adapted with TENT for multiple epochs (5 epochs) involving multiple adaptation steps. Here, the model is encouraged to increase its predictions confidence, which would lead towards trivial or collapsed solutions i.e. predicting a single or set of classes as output irrespective of the input. TENT without diversity regularizer is prone towards trivial solutions and explains the failure in Figure 3 as the split fraction reaches 1.0. TENT+ overcomes the issue and hence the performance is improved. The generalization results shown in TENT paper was mostly conducted in online settings (thus the number of adaptation steps are lower than adapting for multiple epochs) and an offline setting was shown on a comparatively simpler scenario: SVHN to MNIST variants.
>
> **6)What is the justification for the use cases related to data subset experiments?**
> A model trained on samples from a data distribution (e..g. clean data) generalizes well to unseen samples of the similar distribution. Here, the model capture the data distribution sufficiently to generalize to unseen samples. Inspired by this intuition, we study the model generalization when adapting on a fraction of data from an unseen distribution. We found that model generalization to this new distribution is possible after few adaptation steps. On a stream of data with similar distribution, this finding enable the model to adapt on a subset of data and later switch to complete execution mode without adaptation for efficient run time and improved throughput.
>
>  **Summarize results with the mean in Table 1:** We have included mean over the corruptions result in Table1.
>
> **Qualitative results of image transformations:** We present the qualitative results of the input transformation module in Figure A2 (Section A.2).
>
> Lastly, we also incorporated the suggested changes in the text to improve the readability. Thanks for the suggestions.

---

> > ### Comment · Reviewer_9MXJ · 2021-11-30
> > **Thank you for the thorough response and verifications by experiment!**
> >
> > The response addresses all of the points highlighted in my review for rebuttal, and this response and discussion have altered my view of three weaknesses: novelty, restricting the choice of parameters, and the input transformer. I have accordingly raised my score to 6, on the side of acceptance.
> >
> > - Novelty: After reflection and reading the other reviews and responses, I am convinced of the empirical value of this work in evaluating alternative losses, tuned and simplified regularizers, and the use of input transformation for test-time adaptation. While there are clear antecedents in train-time adaptation, or other problem settings, this work reports the experiments for the test-time setting and can guide further improvement.
> > - Parameter Choice: The rebuttal establishes that empirically the proposed losses do not require the deeper layers to be frozen. As the method is not so sensitive to this potential limitation in the experiments, I am not as concerned about it.
> > - Input Transformation: The input transformation step is the most novel part of the method. Additional results were provided by the response and pointed to in the appendix. I still would have liked to see a more thorough analysis of what is learned and how different architectures fare for this purpose, but that can be pursued by future work.
> >
> > The issue with sacrificing accuracy on unshifted data remains, so I cannot raise the score higher.
> >
> > Here are a few last thoughts on points in the rebuttal:
> >
> > > 5)Why does TENT fail as the split fraction reaches 1.0 under data subset experiments? Note that the data subset experiment is an offline evaluation setting where the model is adapted with TENT for multiple epochs (5 epochs) involving multiple adaptation steps.
> >
> > That does explain the difference. I would be more interested in this analysis in the online setting of batch-by-batch prediction and adaptation. That is after all the fully test-time setting, where efficiency of the method is more important for online use, as mentioned in the introduction and related work. The results for offline are informative too, but perhaps could be placed in the supplement.
> >
> > > 6)What is the justification for the use cases related to data subset experiments? [...] On a stream of data with similar distribution, this finding enable the model to adapt on a subset of data and later switch to complete execution mode without adaptation for efficient run time and improved throughput.
> >
> > That is not established here. This result shows the potential for saving computation by adapting to a subset, but without an algorithm to govern _when_ or _when not_ to adapt on the streaming data this improvement in run time and throughput cannot be realized. How is the method to know the data distribution has changed, or that it has sufficiently adapted to it when the data distribution is not changing?

---

### Official Review · Reviewer_nGh4 · 2021-11-02

**Correctness:** 3
**Technical Novelty And Significance:** 2
**Empirical Novelty And Significance:** 3
**Recommendation:** 5
**Confidence:** 4

**Main Review:**

Strength:

The paper introduces an information maximization loss for the test-time BN adaptation on unlabeled target mini-batch data.  The method has been well-motivated by pointing out the limitations in SOTA methods.
- The proposed formulation incorporates the diversity regularizer to avoid the trivial collapsed solutions on mini-batch data.
- It presents the analysis of the gradients of different losses for confidence maximization. And, proposes to use the negative log-likelihood ratio loss [Yao 2020] for TTA  which has non-vanishing gradients for high confidence predictions.
- It proposes to jointly update the IT module and BN parameters for TTA
- The paper provides comprehensive experimental results in the paper and appendix.

Weakness:
- The proposed diversity loss is not very effective on mini-batch data. The results show the improvement is very marginal.
- The paper should provide examples of failure cases,  and more explanation & discussion about the issues in the last paragraph of section 3.
- The novelty of the paper is relatively limited. IM loss, KL based $L_{div}$,  IT module, and the negative log-likelihood ratio are all proposed in the previous works.

**Summary Of The Paper:**

The paper proposes a new loss to improve test-time BN adaptation for domain adaptation. The proposed loss consists of two components: the diversity maximization loss and the confidence maximization loss. Specifically, they use a running estimate for the diversity loss based on KL divergence. They propose the hard and soft likelihood ratio for the confidence loss which has large gradients for high confidence predictions.

**Summary Of The Review:**

Overall, the paper presents an information maximization (IM) loss for TTA on unlabeled target data. My main concerns are that the effectiveness of the proposed $L_{div}$ on mini-batch data, and the scale-normalization problem of the proposed logits. Hopefully, the authors can address my concern in the rebuttal period.

---

> ### Author Response · Authors · 2021-11-22
> **Thank you for providing your valuable feedback. We believe that our responses address reviewer's concerns.**
>
> **1)The proposed diversity loss is not very effective on mini-batch data:**
>     The confidence maximization losses promote high confident predictions but does not prevent the model from predicting only a single class or set of classes independent of the input (also called trivial or collapsed solution). According to Table 1 and Figure 3 from our paper, TENT without the diversity loss falls towards the trivial solutions by dropping its performance during adaptation (e.g. epoch 1 to epoch 5 in Table1) under offline updates and the diversity loss in TENT+ prevent the performance drop and thus stabilize the adaptation. Here the diversity loss improves the performance of TENT by 1\%-5\% for different corruptions and about 19\% for contrast.
>
> **2) Discussion about the last paragraph of section 3:**
>   We mentioned in the paper that the proposed losses could alternatively encourage the network to scale the logits grow larger and larger and still reduce the loss. However, we did not find any considerable differences empirically in the explored settings when adapting the model with or without freezing the top layer. We found that adapting the model with and without freezing the top layers have comparable performance in both online and offline adaptation settings as shown in Table A4 under Section A.3.1 in appendix.  However, we would still recommend freezing the top-most layers as the default choice to be on the safe side.
>
> Note that we used publicly available pretrained models to adapt to distribution shifts and freezing the top-most layers did not require any architectural changes. The main change compared to TENT is that we restrict the choice of adaptable layers a bit further - TENT freezes all but the affine parameters of normalization layers; we freeze some of the top affine layers additionally. Moreover, we  show that the results of TENT+ , HLR and SLR are comparable with and without freezing the top layers of the network. These results indicate that the early layers capture the distribution shift sufficiently to improve the  model adaptation.
>
>  **3)Novelty of the paper:**
> Our approach constitute of input transformation, hard/soft likelihood ratio along with the running estimate of the diversity regularization for test time adaptation. As we point out in the paper, the negative (hard) log-likelihood ratio loss was independently proposed by Yao et al., 2020 (but not the superior soft log-likelihood ratio loss!) as a replacement to the fully-supervised cross entropy loss for the classification task. In Yao et al., 2020, the loss is motivated as follows: "Only the correct class is measured for the cross entropy and it cannot learn to optimize the discrimination between correct class probabilities and the competing ones. Hence, log-likelihood ratio loss is introduced to discriminate correct class from the competing classes". The properties of the loss was not studied/discussed in this prior work and it also not straightforward to recognize the potential of this loss for test time adaptation.
>
> Unlike TENT that uses the well-established entropy minimization, the proposed likelihood ratio losses were never studied for their potential as self-supervised objectives nor used in domain adaptation / test time adaptation problem settings before, to the best of our knowledge. We are the first to identify the advantages of these losses for self-supervised learning and motivate their applicability for test-time adaptation due to their non-saturating gradient property. Besides, soft log-likelihood ratio was introduced in our work (not proposed in Yao et al., 2020) and we show its advantages over other methods for test time adaptation.
>
>  Li et al. arXiv 2020 use a moving average for  diversity regularization but use source data to estimate the gradient of the entropy and Prabhu et al., ICCV 2021 use the last Q=256 instancesto compute the diversity regularizer.  In contrast, our approach does not need source data nor does it need to keep track of the last Q instances. Instead, our work computes moving average over different mini-batches for  diversity regularization where the gradients are estimated using only target data.

---

### Official Review · Reviewer_e5PY · 2021-11-03

**Correctness:** 4
**Technical Novelty And Significance:** 2
**Empirical Novelty And Significance:** 4
**Recommendation:** 6
**Confidence:** 4

**Main Review:**

Strengths
---
The self-supervised log likelihood ratio objective appears novel, as far as I am aware. And, for this problem setting, the combination of the aforementioned techniques is novel and leads to stronger empirical results than what has been previously reported.

The experiments are generally comprehensive and cover, as far as I can see, the important aspects of evaluation. I appreciate the results presented for both the "offline" and "online" adaptation settings, as well as the results adding all of the various techniques to Tent to evaluate whether any technique is of paramount importance.

The paper is generally well written and structured.


Weaknesses
---
I think that the paper has improved on this point, but the motivation behind the general approach is still somewhat shaky. The idea that the model should extract a self-supervised learning signal from data points it is already very confident for still seems strange to me. Imagine if the model was already very confident for the entire batch of data points, but there is a (predicted) class imbalance in the batch. Would it not be the case that the model would adapt in this case when using the proposed approach, even though it would make more sense to not adapt at all, which for example entropy minimization would (roughly) do? And it is in general just unclear to me how incorporating a stronger gradient signal from confident points would help when it comes to ambiguous points. Perhaps what could be useful here is to actually "show this in action", e.g., take a real batch of data during adaptation and demonstrate how the model adapts with the proposed approach vs with Tent. This may provide greater intuition as to why the proposed approach is a good idea.

Negative results are also of interest to the community, and to this end, including results on challenging distribution shift benchmarks such as ImageNet-A and ImageNet-v2, which prior work [1] has shown adaptation to be unhelpful for, would be great. Even just in the appendix, it would still be appreciated.

A final minor nit from my previous review: I would still like to know whether or not a confidence of 0.82 is "low" for the corrupted image datasets or other instances of test distribution shift.

[1] Schneider et al, "Improving robustness against common corruptions by covariate shift adaptation". NeurIPS 2020.

**Summary Of The Paper:**

In the spirit of full disclosure: I have recently reviewed this paper, and several parts of my previous review are still applicable, thus I am copying in these parts when appropriate.

This paper presents a method for test time adaptation based on several techniques. These include a self-supervised adaptation objective based on log likelihood ratios, an additional regularizing objective to encourage diverse predictions, and an input transformation module that is also trained with the aforementioned objectives. Together, these techniques lead to better performance on ImageNet-C and ImageNet-R compared to Tent, a recent and similar test time adaptation method based on entropy minimization.

**Summary Of The Review:**

Primarily due to my concerns above, I am initially recommending a weak accept of this paper. I am happy to engage in discussion with the authors and other reviewers in order to reach a more confident final recommendation.

---

> ### Author Response · Authors · 2021-11-22
> **We thank the reviewer for the positive feedback and believe that our response clears the reviewer's concern.**
>
> **How incorporating a stronger gradient signal from confident points would help?**
> A recent theoretical study [A] has shown (in the fully supervised setting) that the decreasing influence of confident points that are far from the decision boundary and contribute smaller gradients hinders the model representation learning, energy optimization and the growth of margin. Improving the contribution of the confident points has shown to improve the model performance in these three aspects. Our proposed loss also aligns with the similar intuition (but in a self-supervised setting) and encourage the contribution of confident points that further help model performance. We would also like to refer to Appendix A.1 which illustrates the  the benefits of proposed log likelihood ratio loss on a controlled setting.
> Moreover, we conducted an ablation study to show that gradient signal from confident points help model performance. For this, we compute the loss only for the samples with a confidence less than a certain threshold and adapt the model accordingly. We set different confidence threshold [0.6, 0.7, 0.75, 0.8, 0.85, 0.9, 0.95] and adapted the model with SLR and TENT+. The result in tables below  SLR (top) and TENT+ (bottom) showcase the extent to which the gradients from the confident points help to improve the performance with SLR. We see that computing the loss for higher confident samples consistently improved the performance with SLR, whereas the improvements are minor and saturated with TENT+ as the confidence threshold increases. As we stated that entropy have vanishing gradients for higher confidence, the lower confidence range could be seen as less than 0.75 - 0.8 from the results of TENT+.
>
> [A] Zhao, Guangxiang, et al. "Well-classified Examples are Underestimated in Classification with Deep Neural Networks." arXiv preprint arXiv:2110.06537 (2021).
>
> Results of SLR:
>
> |Probability_threshold|Gauss|Glass|Fog|Contrast|
> |:---------:|:---------:|:---------:|:---------:|:---------:|
> |0.6|29.96|30.24|56.94|33.99|
> |0.7|31.43|31.57|57.59|36.78|
> |0.75|32.07|32.24|57.94|38.03|
> |0.8|32.53|32.57|58.75|39.02|
> |0.85|33.15|33.60|58.88|40.35|
> |0.9|33.76|33.98|59.46|41.54|
> |0.95|34.44|34.68|59.98|43.03|
>
> Results of TENT+:
>
> |Probability_threshold|Gauss|Glass|Fog|Contrast|
> |:---------:|:---------:|:---------:|:---------:|:---------:|
> |0.6|27.59|27.23|58.13|23.60|
> |0.7|27.97|27.63|58.34|26.30|
> |0.75|28.21|27.81|58.46|26.74|
> |0.8|28.41|27.95|58.52|27.61|
> |0.85|28.50|28.02|58.61|27.49|
> |0.9|28.57|28.22|58.65|28.12|
> |0.95|28.60|28.30|58.67|28.34|
>
>
> * **Tests on ImageNet-A:**
> We adapted ResNet50 on ImageNet-A dataset and reported the results in Table A12 under section A.8 in appendix as suggested by the reviewer.

---

### Official Review · Reviewer_8M4e · 2021-11-04

**Correctness:** 3
**Technical Novelty And Significance:** 3
**Empirical Novelty And Significance:** 3
**Recommendation:** 6
**Confidence:** 5

**Main Review:**

Strengths

– The paper is mostly well written and easy to follow

– The proposed approach is intuitive, appears effective, and consistently outperforms competing methods for test-time adaptation

– The paper includes a comprehensive set of experiments

– The paper does well to compare with existing entropy minimization alternatives like MaxSquares and Charbonnier penalty. I would recommend including those results and a more detailed discussion in the main paper rather than appendix.

Weaknesses

– The proposed approach is largely a combination of existing methods – TENT (Wang et al., ICLR 2021), negative log-likelihood ratio loss (Yao et al., 2020), and batch-level diversity regularization (Li et al., arXiv 2020, Prabhu et al., ICCV 2021 [A]), for the test-time adaptation setting.

[A] Prabhu, Viraj, et al. "Sentry: Selective entropy optimization via committee consistency for unsupervised domain adaptation." Proceedings of the IEEE/CVF International Conference on Computer Vision. 2021.

– The paper lacks a proper ablation study: while the input transformation module is ablated, what is the individual contribution of each proposed piece?

– The proposed loss does appear to have certain limitations, as it is unbounded and relies on proper scaling via model design (eg. batch norm layers) to prevent logit explosion.

– “the soft likelihood ratio loss creates lower amplitude gradients for low-confidence self-supervision”: this does not appear to match Figure 1 (right), where SLR is slightly larger than HLR for low confidence (<0.2). Further, both SLR and HLR actually appear to have large gradients in this confidence regime as compared to hard pseudolabels – is this not problematic, since that would effectively upweight very low confidence predictions?


----post-rebuttal----

The author response had addressed my concerns about the behavior of the proposed loss. In light of the paper's empirical contributions but limited technical novelty, I recommend a marginal accept.

**Summary Of The Paper:**

Studies the problem of test-time adaptation across distribution shift, and proposes i) a new self-training loss with better stability than entropy minimization ii) using a diversity regularizer and iii) an additional “input transformation” module. The approach is found to lead to improved performance on standard test-time adaptation settings.


**Summary Of The Review:**

Interesting paper on improving test-time adaptation but I have some concerns (see weaknesses). I will be willing to reconsider my rating based on the author response.

---

> ### Author Response · Authors · 2021-11-22
> **Thank you for the valuable feedback. It further improves the understanding of our work.**
>
> **1) The proposed approach is largely a combination of existing methods**
> Note that we have considered TENT and its variant TENT+ as a baseline to our work and our approach is not a combination with TENT. Our approach constitute of input transformation, hard/soft likelihood ratio along with the running estimate of the diversity regularization for test time adaptation. As we point out in the paper, the negative (hard) log-likelihood ratio loss was independently proposed by Yao et al., 2020 (but not the superior soft log-likelihood ratio loss!) as a replacement to the fully-supervised cross entropy loss for the classification task. In Yao et al., 2020, the loss is motivated as follows: "Only the correct class is measured for the cross entropy and it cannot learn to optimize the discrimination between correct class probabilities and the competing ones. Hence, log-likelihood ratio loss is introduced to discriminate correct class from the competing classes". The properties of the loss was not studied/discussed in this prior work and it also not straightforward to recognize the potential of this loss for test time adaptation.
>
> Unlike TENT that uses the well-established entropy minimization, the proposed likelihood ratio losses were never studied for their potential as self-supervised objectives nor used in domain adaptation / test time adaptation problem settings before, to the best of our knowledge. We are the first to identify the advantages of these losses for self-supervised learning and motivate their applicability for test-time adaptation due to their non-saturating gradient property. Besides, soft log-likelihood ratio was introduced in our work (not proposed in Yao et al., 2020) and we show its advantages over other methods for test time adaptation.
>
>  Li et al. arXiv 2020 use a moving average for  diversity regularization but use source data to estimate the gradient of the entropy and Prabhu et al., ICCV 2021 use the last Q=256 instancesto compute the diversity regularizer.  In contrast, our approach does not need source data nor does it need to keep track of the last Q instances. Instead, our work computes moving average over different mini-batches for  diversity regularization where the gradients are estimated using only target data.
>
> **2) What is the individual contribution of each piece in input transformation module?**
> Thanks for suggesting us to perform the ablation study. We have noticed that the  additional channel wise affine transformations didn't bring further consistent improvements and can be ignored from the transformation module. We have included the results from this ablation study in our revised version in Section A.2.1 in appendix.
>
> **3) The proposed loss is unbounded and relies on proper scaling**
> We mentioned in the paper that the proposed losses could alternatively encourage the network to scale the logits grow larger and larger and still reduce the loss. However, we did not find any considerable differences empirically in the explored settings when adapting the model with or without freezing the top layer. We found that adapting the model with and without freezing the top layers have comparable performance in both online and offline adaptation settings as shown in Table A4 under Section A.3.1 in appendix.  However, we would still recommend freezing the top-most layers as the default choice to be on the safe side.
>
> Note that we used publicly available pretrained models to adapt to distribution shifts and freezing the top-most layers did not require any architectural changes. The main change compared to TENT is that we restrict the choice of adaptable layers a bit further - TENT freezes all but the affine parameters of normalization layers; we freeze some of the top affine layers additionally. Moreover, we  show that the results of TENT+ , HLR and SLR are comparable with and without freezing the top layers of the network. These results indicate that the early layers capture the distribution shift sufficiently to improve the  model adaptation.
>
> **4) Soft likelihood ratio loss creates lower amplitude gradients for low-confidence self-supervision**
> We would like to clarify that Figure 1 illustrates a binary classification task. If confidence for class $y1$ is below 0.2, it is above 0.8 for the other class $y2$. Because of this, the model is minimally confident for a confidence of 0.5 for both classes. In this regime, SLR has actually very small gradient amplitude. We will clarify this in our final version.

---

### Author Response · Authors · 2021-11-22
**We thank all the reviewers for providing valuable feedback**

We highly appreciate that you find our work well written, easy to follow, well-motivated with comprehensive and stronger empirical results than prior works [R1, R2, R3], also considering our proposed objective novel [R2] and appreciating further details of our work [R4]. As suggested, we have clarified the details regarding the novelty of our work and also provided additional results to further improve the understanding of our work. We have revised our submission with these additional results. Below we respond to your main criticism and suggestions in detail.

---

### Decision · Program_Chairs · 2022-01-20

**Decision:**

Reject

**Comment:**

The paper considers test time adaptation to distribution shift which is a very important and impactful problem. The authors propose an empirical method that has different pieces, the most important ones being input transformation and confidence maximization and using likelihood ratio loss.

There were various concerns that got addressed during the rebuttal period such as, novelty of the proposed method, ablation study of different parts of the model, novelty and importance of diversity regularizer, choice of optimization. However there are still three remaining concerns that addressing them will improve the paper significantly: First, clear motivation behind the method for the cases when the model is certain but we have data imbalance. Second, analysis in the online setting of batch-by-batch prediction and adaptation. Third,
establishing the claim regarding data subset experiment that it enable the model to adapt on a subset of data and later switch to complete execution mode without adaptation for efficient run time and improved throughput. How is the method to know the data distribution has changed, or that it has sufficiently adapted to it when the data distribution is not changing?